# Forcing the SURFEX/Crocus snow model with combined hourly meteorological forecasts and gridded observations in southern Norway

Hanneke Luijting[1], Dagrun Vikhamar-Schuler[1], Trygve Aspelien[1], Åsmund Bakketun[1], and Mariken Homleid[1]

[1]The Norwegian Meteorological Institute, PO Box 43 Blindern, 0313 Oslo, Norway

*Correspondence to:* Hanneke Luijting (hanneke.luijting@gmail.com) or Dagrun Vikhamar-Schuler (dagrunvs@met.no)

**Abstract.** In Norway, thirty percent of the annual precipitation falls as snow. Knowledge of the snow reservoir is therefore important for energy production and water resource management. The land surface model SURFEX with the detailed snowpack scheme Crocus (SURFEX/Crocus) has been run with a grid spacing of 1 km over an area in southern Norway for two years (01 September 2014 - 31 August 2016). Experiments were carried out using two different forcing data sets: 1) hourly forecasts from the operational weather forecast model AROME MetCoOp (2.5 km grid spacing) including post-processed temperature (500 m grid spacing) and wind, and 2) gridded hourly observations of temperature and precipitation (1 km grid spacing) combined with meteorological forecasts from AROME MetCoOp for the remaining weather variables required by SURFEX/Crocus. We present an evaluation of the modeled snow depth and snow cover, as compared to 30 point observations of snow depth and to MODIS satellite images of the snow-covered area. The evaluation focuses on snow accumulation and snow melt. Both experiments are capable of simulating the snow pack over the two winter seasons, but there is an overestimation of snow depth when using meteorological forecasts from AROME MetCoOp (bias of 20 cm and RMSE of 56 cm), although the snow-covered area throughout the melt season is better represented by this experiment. The errors, when using AROME MetCoOp as forcing, accumulate over the snow season. When using gridded observations, the simulation of snow depth is significantly improved (the bias for this experiment is 7 cm and RMSE 28 cm), but the spatial snow cover distribution is not well captured during the melting season. Underestimation of snow depth at high elevations (due to the low elevation bias in the gridded observation dataset) is likely causing the snow cover to decrease too soon during the melt season, leading to unrealistically little snow by the end of the season. Our results show that forcing data consisting of post-processed NWP data (observations assimilated into the raw NWP weather predictions) are most promising for snow simulations, when larger regions are evaluated. Post-processed NWP data provide a more representative spatial representation for both high mountains and lowlands, compared to interpolated observations. There is however an underestimation of snow ablation in both experiments. This is generally due to the absence of wind-induced erosion of snow in the SURFEX/Crocus model, underestimated snow melt and biases in the forcing data.

# 1 Introduction

Snow is a key element in the hydrological cycle. Seasonal snow covers large areas of the Northern Hemisphere and the Arctic. In these areas the snow cover extent in spring has reduced more rapidly the past 40 years than over the past 90 years (Brown and Robinson, 2011; Brown et al., 2017). The largest declines in snow cover extent and duration are observed in Arctic coastal areas, e.g. in Scandinavia (Rasmus et al., 2015; Brown et al., 2017). In Norway there is a general trend towards a later start and an earlier end of the snow season, although there are large annual variabilities (Hanssen-Bauer et al., 2015, 2017). Trends in snow depth may vary with elevation, as observed for some Norwegian regions (Skaugen et al., 2012; Dyrrdal et al., 2013). Information about seasonal changes in snow duration and amounts are important for many societal applications and for Arctic ecosystem changes. An overview of changes in snow and impacts due to these changes, is provided by Bokhorst et al. (2016).

In Norway 30% of the annual precipitation falls as snow (Saloranta, 2012). Observations show that changes in the winter climate over the past 50 years, and particularly since 2000, give more winter warming and rainfall events (Vikhamar-Schuler et al., 2016; Kivinen et al., 2017). This, in turn, affects the internal snow structure giving e.g. more wet snow conditions and ground-ice layering (Johansson et al., 2011; Vikhamar-Schuler et al., 2013). Updated information on the daily local snow properties (e.g. depth, SWE, density profile, crystal structure etc) and snowmelt in mountainous and lowland areas is very useful for many applications, notably local flood prediction, hydropower production planning, snow avalanche prediction, tourism and traffic flow management. Typical information needed for these applications are daily forecasts of snow properties and snow melt (for the next days), but depending on the application, also knowledge of snow conditions for the past winter(s), last month(s), last week(s), past 3 days and yesterday.

A wide range of empirically and physically based snow models have been developed and reported in the literature, see e.g. Magnusson et al. (2015). Models differ in several ways e.g. the parameterization and simplification of snow processes, the spatial and temporal resolution or the need for input data of various weather elements. The need for input data is therefore usually larger and more detailed for physically based models than for the empirically based models. Empirical models often need calibration. Several snow model intercomparison projects have also been performed, e.g. Etchevers et al. (2004) and Essery et al. (2013). These studies show that no single model always performs best, and there is no clear link between model complexity and performance. However, physically-based models, which includes prognostics of snow density and albedo, tend to perform better (Essery et al., 2013).

In Norway, both the national operational flood forecasting and hydropower companies use the HBV model for hydrological forecasting, which includes an empirical degree day model for snow simulations (Bergstrøm, 1976; Sælthun, 1996; Ruan and Langsholt, 2017). Snow maps of depth, water equivalent, snow melt, snow wetness and skiing conditions are also produced operationally on a daily basis and published at www.seNorge.no and www.xgeo.no (Saloranta, 2016), and these maps are used by the national snow avalanche service (Barfod et al., 2013; Engeset, 2013). Both these applications use gridded near real-time observations of temperature and precipitation (Mohr, 2008; Lussana et al., 2018a).

Another type of forcing data for snow models are numerical weather forecasts (NWP) from atmospheric models. NWP data provides all the basic environmental variables required by physically-based snowpack models at hourly time steps (e.g. air

temperature, relative humidity, wind speed, precipitation rate, incoming short and longwave radiation). Using sub-daily (e.g. hourly) data in snow modeling should contribute to improved representation of snow melt processes (e.g. diurnal freeze/thaw cycles) and precipitation phase. SNOWPACK (Bartelt and Lehning, 2002; Lehning et al., 2002), SURFEX/ISBA/Crocus (Vionnet et al., 2012) and JULES (Best et al., 2011) are examples of models with multi-layer snow schemes of different complexity

aiming to simulate the surface energy balance and the internal layering of the snowpack. SNOWPACK and Crocus are used in the operational snow avalanche service in Switzerland and France, respectively (Fierz et al., 2013; Lafaysse et al., 2017). Many studies show how high resolution NWP data are very valuable in driving these snow models, see e.g. Bellaire et al. (2011, 2013); Horton et al. (2015); Vionnet et al. (2016); Quéno et al. (2016). NWP data have also been used as driving data in hydrometeorological models (Carrera et al., 2010, e.g.).

For a point location in the Columbia Mountains, Western Canada, Bellaire et al. (2011, 2013) used 15 km resolution weather forecasts from the NWP Global Environmental Multiscale Model (GEM) to force the SNOWPACK model. This study was later extended to a gridded area in the same region by Horton et al. (2015) who forced the SNOWPACK model using 2.5 km resolution NWP data from the Limited Area Model version of GEM (GEM-LAM) model. The use of NWP data as precipitation forcing for snow models was analysed and discussed by Schirmer and Jamieson (2015). They compared two NWP datasets

(GEM: 15 km and GEM-LAM: 2.5 km spatial resolution) over complex mountainous terrain during winter time, and found that the highest resolution dataset performed best in terms of precipitation forecasts. Bernier et al. (2011) used a downscaling technique to account for local terrain effects on the surface temperatures not resolved by the low-resolution NWP model. With higher spatial resolution of the NWP models, the orographic precipitation is better reproduced. In the French Alps, high-resolution forecasts (2.5 km) from the AROME NWP model were used to drive Crocus (Vionnet et al., 2016). A similar study

using the same AROME NWP model was carried out for the French and the Spanish Pyrenees by Quéno et al. (2016). Both these studies showed that high-resolution NWP data represents a very useful and promising data source for snow models to produce snow maps. However, the authors point out some limitations of using only NWP data. Terrain effects are not well enough accounted for on a kilometric scale, whereas future development of sub-kilometric scale NWP data might improve e.g. terrain effects on the incoming solar radiation. Combining NWP data with other data sources (e.g. observations, radar)

might improve the forcing data, particularly the precipitation fields. Redistribution of snow due to wind is another difficult issue in mountainous areas. Running snow models with ensemble based forecasts is a promising method to account for these uncertainties (Vernay et al., 2015b; Lafaysse et al., 2017).

Weather forecasting models are presently evolving fast, and they include more and more detailed parametrization of land surface processes connected to snow and soil. SURFEX (Surface Externaliseé) (Masson et al., 2013) is an example of a land

surface model, which can be run both inline as part of an atmospheric weather forecast model, for example AROME MetCoOp (Müller et al., 2017), or offline as a stand-alone model. At the Norwegian Meteorological Institute (MET Norway), the AROME MetCoOp model is run operationally to provide short-term weather forecasts covering large parts of the Nordic region (Müller et al., 2017).

Data assimilation methodologies for snow are presently likewise evolving fast, see e.g. Carrera et al. (2015). Snow analysis

incorporating in situ observations, satellite data, and estimates from NWP-driven physical snowpack models is an alternative to

NWP-driven offline runs. In Norway, observed snow depth at weather stations are daily assimilated into the AROME MetCoOp model, in order to improve the predicted surface air temperature. However, the MetCoOp model uses the most simple snow scheme (D95, Douville et al. (1995)) of the three snow schemes implemented in the SURFEX model (Boone and Etchevers, 2001; Masson et al., 2013). The D95 scheme models the snowpack as a single layer, with two prognostic variables: SWE and snow density. For many of the above mentioned applications, information of other snow properties such as internal layering as well as the density, grain size, temperature, wetness etc. of the different layers are of high interest. This is not provided by the D95 snow scheme.

In this study we therefore evaluate the performance of the SURFEX model, using the Crocus snow scheme (Vionnet et al., 2012) for Norwegian snow conditions. Crocus is the most advanced snow scheme implemented in the SURFEX model (Boone and Etchevers, 2001; Masson et al., 2013). SURFEX/Crocus has not previously been run in a gridded stand alone version for regions in Norway (as a 2D study). However, the model has earlier been tested for single points (1D study) with observations from weather stations and NWP data (Vikhamar-Schuler et al., 2011). Our study is carried out as part of several research projects within hydropower and flood forecasting. The domain was chosen to cover mountains in southern Norway and to include a cross-section from west to east that crosses the watershed in this region. This domain includes catchment areas that are of high interest to hydropower companies.

The aim of our study is to test the performance and the benefit of different gridded forcing datasets as input to the SUR-FEX/Crocus model, and validate the simulated snow amounts and snow melt patterns in the selected domain. The originality of our work is linked to the unique combination of using both raw weather predictions, post-processed weather predictions and gridded observations, which we expect should provide an improved performance of the snow simulations compared to e.g. using only raw weather predictions. Combining observations and NWP data for important weather variables (temperature, precipitation and precipitation phase) as driving data for the snow simulations should better represent the actual observed weather conditions. Experiments were performed by applying two different data sets from the winters 2014/2015 and 2015/1016 as forcing to the SURFEX/Crocus model: 1) Predictions from the AROME MetCoOp model with a grid spacing of 2.5 km (Müller et al., 2017), where both the temperature and the wind data were improved by post-processing algorithms, and 2) Gridded observations of precipitation and temperature (GridObs) with a grid spacing of 1 km (Lussana et al., 2018b, a). Both data sets have hourly temporal resolution, and are discussed in detail in section 2.3.

Although AROME-SURFEX/Crocus has previously been used over the southern European mountain chains in the French Alps (46°N, 9°E) and the French/Spanish Pyrenees (42°N, 1°E) (Vionnet et al., 2016; Quéno et al., 2016), neither of our two datasets described above have been used as forcing for SURFEX/Crocus for Norwegian mountains and lowland regions before. Our study area is located in Northern Europe at 61°N, 8°E, which is at least 15 degrees further north. According to the Köppen-Geiger climate classification system (Köppen, 1936), the climate in South-Norway is different from the Pyrenees, while the climate classes are partly the same for South-Norway and the Alps. However, these coarse climate classes generally account for average temperature and precipitation in an area, and do not fully account for the differences in probability distribution functions describing the regional climate variability (e.g. precipitation intensities, frequencies and extremes) in these mountainous areas. A west-east transect crossing the mountain chain in South Norway comprises a climatic transect from maritime,

alpine to more continental climate. Snow conditions and stratigraphy vary regionally as outlined by e.g. Sturm et al. (1995), who defined six snow classes, of which at least two classes are inside our domain (maritime and alpine). The SURFEX/Crocus snow model may therefore perform differently in individual regions. Our study contributes to a development which can produce new supplementary snow information (including snow stratigraphy) and thereby may contribute to the development of

a future system for daily snow mapping. The performance of the SURFEX/Crocus model is also compared with three other snow models including the seNorge model in a separate study by Skaugen et al. (2018), a study which also shows that there is no "best" snow model.

## 2   Model setup and data sets

### 2.1   The SURFEX/Crocus model

The model used in this study is the detailed snowpack model Crocus (Brun et al., 1992; Vionnet et al., 2012) coupled with the ISBA land surface model within the SURFEX (Surface Externaliseé) interface (Masson et al., 2013). We applied the ISBA-DIF multi-layer soil scheme (Boone et al., 2000; Habets et al., 2003), which uses a diffusive approach for modeling the heat and moisture transport in the soil. The soil was divided into 14 layers, of which the thickness of the individual layers increases with the soil depth. The bottom depth of the lowest layer was 12 m. The HSWD (Harmonized World Soil Database) 1 km resolution

database for soil texture (FAO/IIASA/ISRIC/ISS-CAS/JRC, 2012) was used for the soil properties.

    The snowpack scheme Crocus models the physical properties of up to 50 dynamic layers within the snowpack, as well as the underlying ground. Once the snowpack reaches a threshold of 1 kg m$^{-2}$ SWE, the fractional snow cover over a grid point is assumed to be 1. The SURFEX/Crocus model can be run in stand-alone (or offline) mode, or fully coupled to an atmospheric model.

For this study, the SURFEX/Crocus model was used in an offline mode, on a 0.01° grid (approximately 1 km), with a 5 minute internal time step and output every hour. The orography was taken from GTOPO30 global digital elevation model (DEM) from the U.S. Geological Survey, which has a grid spacing of 1 km. The transport of snow by wind is not simulated. The SURFEX/Crocus model was run for two winter seasons: from 1 September 2014 until 31 August 2016. These dates were chosen because the hydrological year starts on 1 September, and at that time there is normally no snow in the mountains. In

this study, we start a new simulation on 1 September, with no snow present, and with default values for soil properties, for both 2014/2015 and 2015/2016. The default soil temperature is 11.9 °C for the uppermost surface soil layer for 0 m.a.s.l. (sea-level height). The soil temperature is reduced with increasing terrain elevation using a lapse rate of 0.65 °C per 100 m, leading to a surface soil temperature of 1.3 °C at 1000 m.a.s.l.. We estimate these surface soil values to be representative of the September climate in our study area. Higher temperatures in the deepest soil layers may however represent an uncertain heat contribution

for the snow modeling. This effect should be similar for all the experiments though, since the initialization is the same.

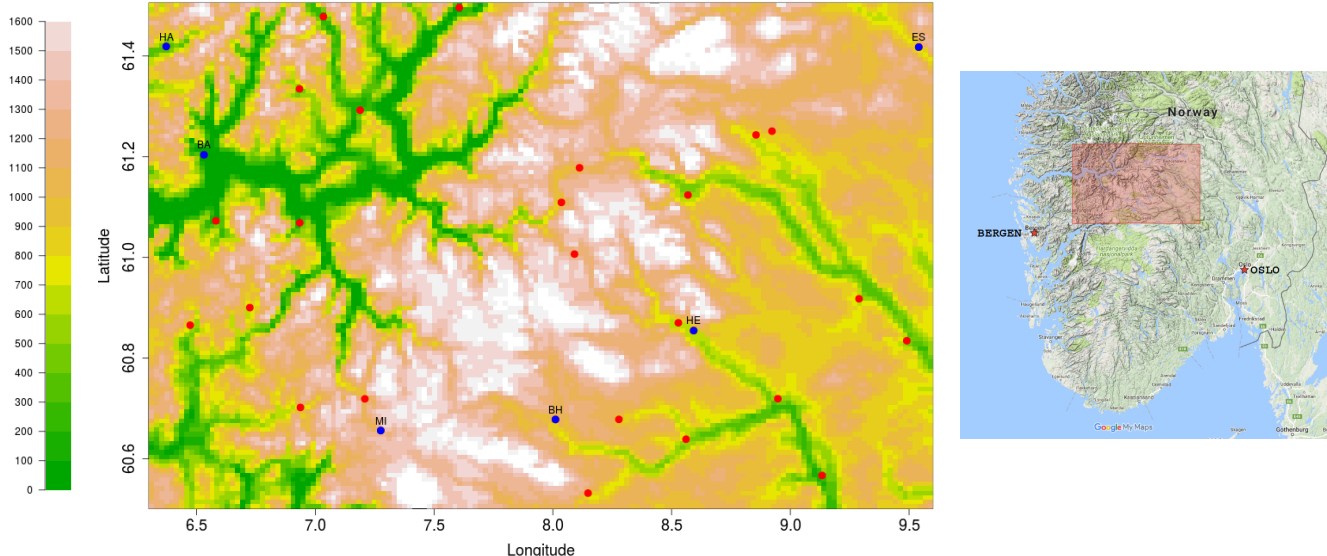

**Figure 1.** Map showing the domain over which the SURFEX/Crocus model was run on the right (map data: Google), with on the left a map showing the elevation over the SURFEX/Crocus model domain, and the locations of the 30 observations used in this paper (indicated by blue and red dots). The blue dots indicate the 6 stations used in Fig. 4: BA = Balestrand Brannstasjon, HA = Haukedal, HE = Hemsedal II, ES = Espedalen, BH = Bakko i Hol and MI = Midtstova.

## 2.2 The study area

Figure 1 shows the domain over which the SURFEX/Crocus model was run, and the elevation over the model domain. The domain covers nearly 20.000 km$^2$ (111 x 175 km), and contains 100 x 330 grid points. As mentioned in the introduction, the study area was chosen to cover the mountains in southern Norway and to include a cross-section from west to east that crosses
the watershed in this region, as well as to include several catchment areas that are of interest to hydropower companies. The domain covers elevations from 0 masl. along fjords up to the highest mountain in Norway (2468 masl.). Therefore, the area includes different vegetation zones, ranging from high mountains above the tree line, sparsely forested and densely forested areas. This makes it a challenging area for snow modeling.

Due to the watershed and the prevailing weather patterns, there is a large gradient in precipitation amount over the domain.
The far western parts of the domain receive on average around 1500 mm of precipitation during a winter season, while the eastern parts only receive 100-300 mm (Hanssen-Bauer et al., 2015). The western part of the domain has a maritime climate while the eastern part has a more inland climate, which means the average temperature during winter is higher at the western part of the domain (around or just below 0 °Celsius), compared to the eastern side (around -10 °Celsius) (Hanssen-Bauer et al., 2015). This means the gradient in average snowfall amount is not as large as the gradient in precipitation amount, but the
western part of the domain still receives significantly more snow than the eastern part (Hanssen-Bauer et al., 2015).

| | AROME-Crocus | GridObs-Crocus |
|---|---|---|
| Air temperature [K] | AROME-MetCoOp post-processed | Gridded observations |
| Specific humidity [kg kg$^{-1}$] | AROME-MetCoOp | AROME-MetCoOp |
| Wind speed [m s$^{-1}$] | AROME-MetCoOp post-processed | AROME-MetCoOp |
| Wind direction [degrees] | AROME-MetCoOp | AROME-MetCoOp |
| Incoming direct shortwave radiation [W m$^{-2}$] | AROME-MetCoOp | AROME-MetCoOp |
| Incoming longwave radiation [W m$^{-2}$] | AROME-MetCoOp | AROME-MetCoOp |
| Surface pressure [Pa] | AROME-MetCoOp | AROME-MetCoOp |
| Rainfall rate [kg m$^{-2}$ s$^{-1}$] | AROME-MetCoOp post-processed[1] | Gridded observations[2] |
| Snowfall rate [kg m$^{-2}$ s$^{-1}$] | AROME-MetCoOp post-processed[1] | Gridded observations[2] |

**Table 1.** Description of the forcing data sets used in the two experiments: 1) AROME-Crocus; and 2) GridObs-Crocus. The rainfall rate and snowfall rate have been derived from the total precipitation by using a threshold temperature of +0.5 °Celsius, using the temperature from [1]: the post-processed AROME-MetCoOp temperature and [2]: the gridded observations of temperature.

## 2.3 Forcing data sets

The SURFEX/Crocus model requires atmospheric forcing. For this study, we have used two different sets of forcing data. Table 1 shows an overview of which variables the SURFEX/Crocus model requires and the different sources used in the two experiments: 1) AROME-Crocus and; 2) GridObs-Crocus. AROME-Crocus uses both raw and post-processed forecasts from the AROME MetCoOp model (described below in section 2.3.1) while GridObs-Crocus uses a combination of gridded observations of precipitation and temperature, described in section 2.3.2. All forcing data have hourly temporal resolution.

### 2.3.1 Numerical weather forecasts (AROME-MetCoOp)

AROME MetCoOp is a high-resolution, non-hydrostatic, convective-scale weather prediction model operated by a bilateral cooperative effort [Meteorological Cooperation on Operational Numerical Weather Prediction (MetCoOp)] between the Norwegian Meteorological Institute and the Swedish Meteorological and Hydrological Institute (Müller et al., 2017), operational since March 2014. The core of the model is based on the convection-permitting Applications of Research to Operations at Mesoscale (AROME) model developed by Météo-France (Seity et al., 2011). It has been modified and updated to suit advanced high-resolution weather forecasts over the Nordic regions, see Müller et al. (2017) for details. The horizontal grid spacing is 2.5 km and the domain covers the Nordic countries. The atmosphere is divided into 65 vertical levels, with the first level at approximately 12.5 m height. Atmosphere–surface interactions and surface–soil processes are described by SURFEX (Masson et al. 2013). The fluxes computed by SURFEX at the atmosphere–surface interface serve as the lower boundary conditions for the atmosphere within AROME MetCoOp. All surface processes are treated as one-dimensional vertical processes.

AROME MetCoOp operates with a 3-hourly update cycling, where initial fields of atmospheric and land surface variables are corrected with observations through data assimilation. Observations of air temperature, relative air humidity and snow

depth are used in the surface analysis (Müller et al., 2017). At every main cycle (0000, 0600, 1200, and 1800 UTC) a 66-h forecast is produced. Forcing for our study is taken from the 4 main cycles, with successive 3-8h lead time (0-8h lead time for the 0000 UTC cycle, and 3-5h lead time for the 1800 UTC cycle) forecasts combined into a forcing file for each day. These lead times were chosen to avoid the first hours of a cycle when the model might have spin-up issues, and to make use of all available cycles with the shortest possible lead time (model error increases with lead time, see for example Homleid and Tveter (2016)).

For temperature and wind speed we used statistically post-processed AROME-MetCoOp forecasts to force SURFEX/Crocus, described by Køltzow (2017). These post-processed weather variables are produced operationally by MET Norway for the weather forecast website YR (https://www.yr.no/). The temperature grid has a spatial resolution of 500 m, and is produced using a Kalman filter correction at observation stations (Homleid, 1995). Horizontal interpolation is carried out using decreasing weights with increasing distance from the station. The temperature is further corrected for terrain elevation, which also takes into account vertical temperature profiles in inversion situations in winter time. The AROME-MetCoOp wind speed was statistically post-processed to represent the maximum wind speed at 10 m during the last hour. In addition, correction factors are applied to the wind speed depending on wind direction and region (Køltzow, 2017). The other variables from the raw AROME-MetCoOp 2.5 km forecasts were interpolated to 1 km spatial resolution using bilinear interpolation, in order to combine the meteorological forecasts with the gridded observations (with a spatial resolution of 1 km) and to run the SUR-FEX/Crocus model with 1 km grid spacing. The 500 m post-processed AROME-MetCoOp temperature (Køltzow, 2017) was also interpolated by a bilinear method to 1 km resolution. The spatial interpolation was carried out using the File Interpolation, Manipulation and EXtraction library (http://fimex.met.no).

SURFEX/Crocus requires a separate snowfall and rainfall rate. A threshold temperature of +0.5°Celsius was applied for determining snowfall or rainfall. This threshold temperature is commonly used for hydrological purposes in Norway (see for example Skaugen (1998)). The post-processed AROME-MetCoOp temperature was used to compute precipitation phase based on the total precipitation predicted from the AROME-MetCoOp model. Correct precipitation phase estimation is crucial for good snow simulations. An additional test was carried out on estimating precipitation phase using the raw AROME-MetCoOp snowfall and rainfall at 2.5 km resolution. AROME's own microphysics should provide good precipitation phase estimates, but these are most representative at 2.5 km spatial resolution on the model's own terrain height. In our study area, the terrain variability is very large, particularly in the western regions where the terrain often rises from 0 masl. at the fjords to more than 1000 masl. over very short distances. In these kind of areas, terrain-adjusted precipitation phase determination is necessary. The impact on the snow simulations of using these two different ways of estimating precipitation phase are discussed in Section 4.2.

### 2.3.2 Gridded observations (GridObs)

In an earlier study by Vikhamar-Schuler et al. (2011) , it was shown that snow modeling with the SURFEX/Crocus model has highest sensitivity to the temperature and precipitation input datasets. Best results were obtained when the model was forced

with observations of temperature and precipitation, while replacing other input parameters with meteorological forecast data did not increase errors notably.

Hourly gridded observations of temperature and precipitation are available on a 1 km grid over Norway. This dataset uses all measurements available in MET Norway's Climate database (Frost, 2018). The station distribution is uneven, with more stations in the southern part of Norway and a sparser network in the north and in the mountains. There is a low elevation bias, where most stations are located at lower elevations (e.g. valley bottoms) and few stations are located above 1000 m.a.s.l. (Lussana et al., 2018a). The hourly precipitation values have been obtained by using a two-step procedure. The spatial interpolation method described by Lussana et al. (2018a) has been applied independently to daily and hourly precipitation totals. This method is built on classical methods (such as optimal interpolation and successive-correction schemes) and (spatial) scale-separation using geographical coordinates and elevation as complimentary information in the interpolation. It is based on iterating a statistical interpolation scheme over a decreasing sequence of spatial scales, from synoptic to kilometer scale. Further details can be found in Lussana et al. (2018a). The daily precipitation totals have been disaggregated to an hourly time with a procedure similar to the one described by Vormoor and Skaugen (2013). The two-step procedure has been implemented so that the final hourly product can benefit from the more accurate daily quantitative estimates that are based on a denser network of stations, if compared to the hourly ones. The method to obtain hourly temperature values is described in Lussana et al. (2018b), while the resulting temperature dataset is described and evaluated in Lussana et al. (2016).

The resulting gridded temperature dataset can be regarded as an unbiased estimate of the true temperature both at grid points and at station locations. Only for the most extreme negative values (temperatures below -30 °Celsius) there is a systematic warm bias of about 1 °Celsius (Lussana et al., 2016, 2018b). For precipitation, Lussana et al. (2018a) found that the precision of the estimates (at grid points) is about ±20%, but there is a systematic underestimation of precipitation in data-sparse areas and for intense precipitation.

The first version of these gridded data sets (called seNorge v1.0) included a correction factor for precipitation at individual stations due to undercatch (Lussana et al., 2018a). However, these correction factors were evaluated and found unrealistic in the mountains, giving too much precipitation (Saloranta, 2012). In the next version of these gridded datasets (seNorge v.2.0), all undercatch correction factors were removed, and interpolation was based on uncorrected precipitation. For the same reason, the hourly gridded dataset used in this study does not contain any correction for precipitation undercatch.

The number of stations included in the gridded dataset varies over time. The numbers of stations within the SURFEX/Crocus domain are: 20-30 stations for hourly precipitation, 90-100 stations for daily precipitation and 70-100 stations for temperature. Stations just outside the domain are included in this estimate as they are used in the interpolation and are therefore part of the gridded dataset used in this study.

Snowfall and rainfall rate was estimated assuming rain/snow separation at +0.5 ° Celsius (using the gridded observations of temperature available on the same grid), the same threshold as used in the AROME-MetCoOp forcing dataset.

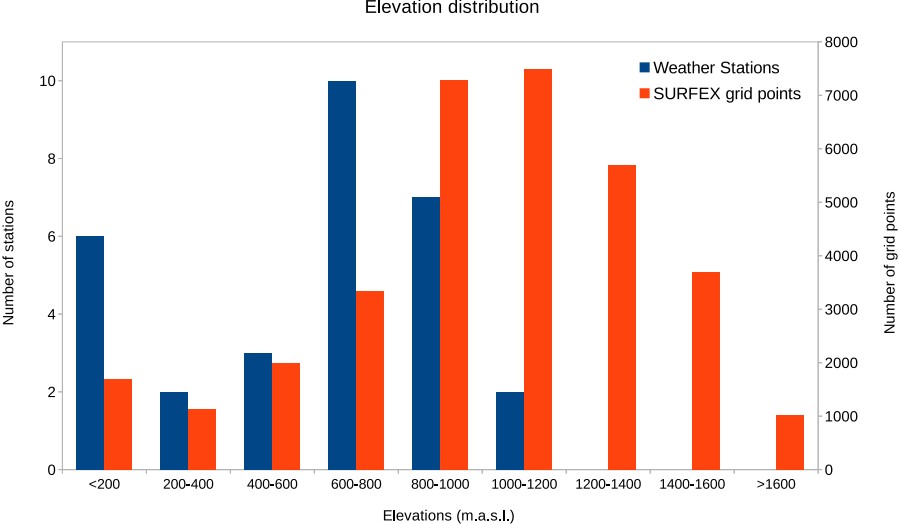

**Figure 2.** Distribution of elevation for the 30 snow depth stations used in this study (in blue, on left axis), and of the grid points in the SURFEX domain (in red, on right axis).

## 2.4 Validation data set

We use two different data sets to validate the results from both experiments: point observations of snow depth and snow cover maps derived from MODIS satellite images.

### 2.4.1 Snow depth observations

Observations of daily snow depth from 30 stations were selected for verification of the model results (see Fig. 1 for their locations within the domain). Nearly all stations (25 out of 30) are official meteorological stations run by the Norwegian Meteorological Institute, while a few stations are owned by other institutions (municipalities, energy producing companies and Bane Nor, the state-owned company responsible for the Norwegian national railway infrastructure). Data from all stations are freely available from the climate database of the Norwegian Meteorological Institute (Frost, 2018). All stations measure daily
snow depth, nearly all (29 out of 30) measure precipitation, and 9 stations also measure air temperature. The stations were selected based on the availability of snow depth observations between 1 September 2014 and 31 August 2016. The locations of the stations are reasonably well distributed over the domain (see Fig. 1) and their elevations range between 14 and 1162 meters above sea level. Figure 2 shows the elevation distribution of all stations used in this study. Along with the distribution of elevations of grid points in the SURFEX domain, Fig. 2 shows a typical issue of low elevation bias in the observing network of
the Norwegian Meteorological Institute, also illustrated in Lussana et al. (2018a). Stations are located at elevations that seldom exceed 1000 m.

A nearest neighbor method was used to evaluate the SURFEX/Crocus experiments with the surface snow depth observations. In a domain with deep valleys and high mountains, it is difficult to match the exact elevation of the weather stations with the nearest grid point in the SURFEX/Crocus experiments. As there were only 30 stations with high quality snow depth observations in the domain, it was decided not to filter out stations based on these elevation differences. The influence of
elevation differences in the evaluation is discussed in section 4.1.

Daily snow depth observations taken at 06 UTC have been used for direct comparison to snow depth from the SURFEX/Crocus simulations. The observations were also used to calculate the start, length and end of the snow season, to compare against model results. The length of the snow season is defined as the number of days with more than 5 cm snow during a year. The 5 cm threshold was also used by Vionnet et al. (2016), although they used continuous snow on the ground as an additional
condition. The start of the snow season is defined as the first day with more than 5 cm of snow, and the end of the snow season as the day after the last day with more than 5 cm of snow.

### 2.4.2   MODIS snow cover images

MODIS (Moderate Resolution Imaging Spectroradiometer; http://modis.gsfc.nasa.gov/) snow cover images (Hall and Riggs, 2007; Klein and Stroeve, 2002) with a resolution of 500 m were available and processed for the melt season of the 2014-2015
winter. The same method as described by Lussana et al. (2018a) and Skaugen et al. (2018) was used to obtain estimates of the daily snow-cover extent over the domain: the MODIS images were converted to snow-covered area (SCA) on a scale from 0-100% coverage using a method based on the Norwegian linear reflectance to snow cover algorithm (NLR) (Solberg et al., 2006). The input to the NLR algorithm is the normalized difference snow index signal (NDSI- signal) (Salomonson and Appel, 2004).
The MODIS images were used for visual and quantitative comparison of the snow melt pattern from satellite images and from both SURFEX/Crocus experiments. For this purpose, four dates with cloud free conditions were selected throughout the melt season: 15 March 2015, 20 April 2015, 15 May 2015 and 04 July 2015.

## 3   Results

### 3.1   Snow depth

A density scatter plot of daily observed and simulated snow depth for both experiments and the two winter seasons 2014/15 and 2015/16 is shown in Fig. 3. Zero snow depth pairs were excluded. GridObs-Crocus is in reasonably good agreement with the observations ($R^2$=0.78), although there are cases of over- and underestimation of around 100 cm, while AROME-Crocus shows significantly more variability and overestimation of snow depth ($R^2$=0.52). To investigate the snow depth at individual stations over a range of station altitudes in more detail, Fig. 4 shows snow depth plots for six locations: two located below 400
m, two located between 500 and 900 m, and two above 900 m (which in our study area means they are located above the tree line). For the location of these six stations within the domain, see Fig. 1 in which they are indicated with blue dots. These six

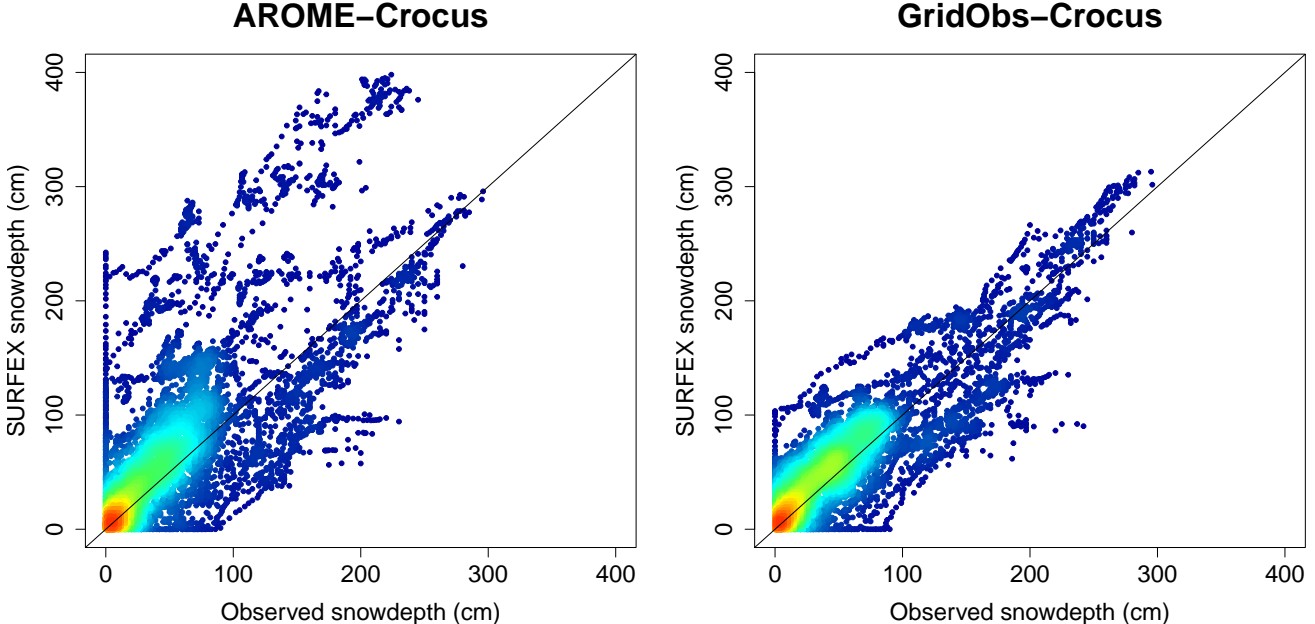

**Figure 3.** Scatter density plot of daily observed and simulated snow depth (cm) for AROME-Crocus (left) and for GridObs-Crocus (right) for the 30 snow depth stations, from 01 September 2014 to 31 August 2016. The density ranges from low in blue to high in red.

stations show that AROME-Crocus overestimates the snow depth for the highest altitudes (Bakko i Hol and Midtstova, which are both situated above 900 masl.), while it underestimates the snow depth for the lowest stations (Balestrand and Haukedal). The two lowest stations are located in the western part of the study area where terrain gradients are very steep.

The snow depth from GridObs-Crocus is closer to the observed snow depth, but at times underestimates the snow depth
(most notably for the first winter season at Haukedal (329 masl.) and Midtstova (1162 masl.)). Episodes when the snow depth decreases during the winter season (apart from snow melt in spring) are not always well captured by the SURFEX/Crocus experiments, and this issue is partly responsible for the overestimation of snow depth.

The results for Hemsedal II (604 masl., see Fig. 4) are of particular interest, as this is the only station measuring snow depth
but not precipitation (and therefore not part of the gridded observation dataset used as input for GridObs-Crocus). GridObs-Crocus overestimates the snow depth at Hemsedal II, but slightly less than AROME-Crocus does. The bias in snow depth at Hemsedal II for the two seasons combined is 25 cm for GridObs-Crocus and 29 cm for AROME-Crocus. When compared to the bias (7 cm for GridObs-Crocus and 20 cm for AROME-Crocus) for all stations for the two seasons combined, it shows that Hemsedal II performs slightly worse than most stations in AROME-Crocus. For GridObs-Crocus, the bias at Hemsedal II is
significantly larger than for most other stations. The fact that GridObs-Crocus outperforms AROME-Crocus even at a station

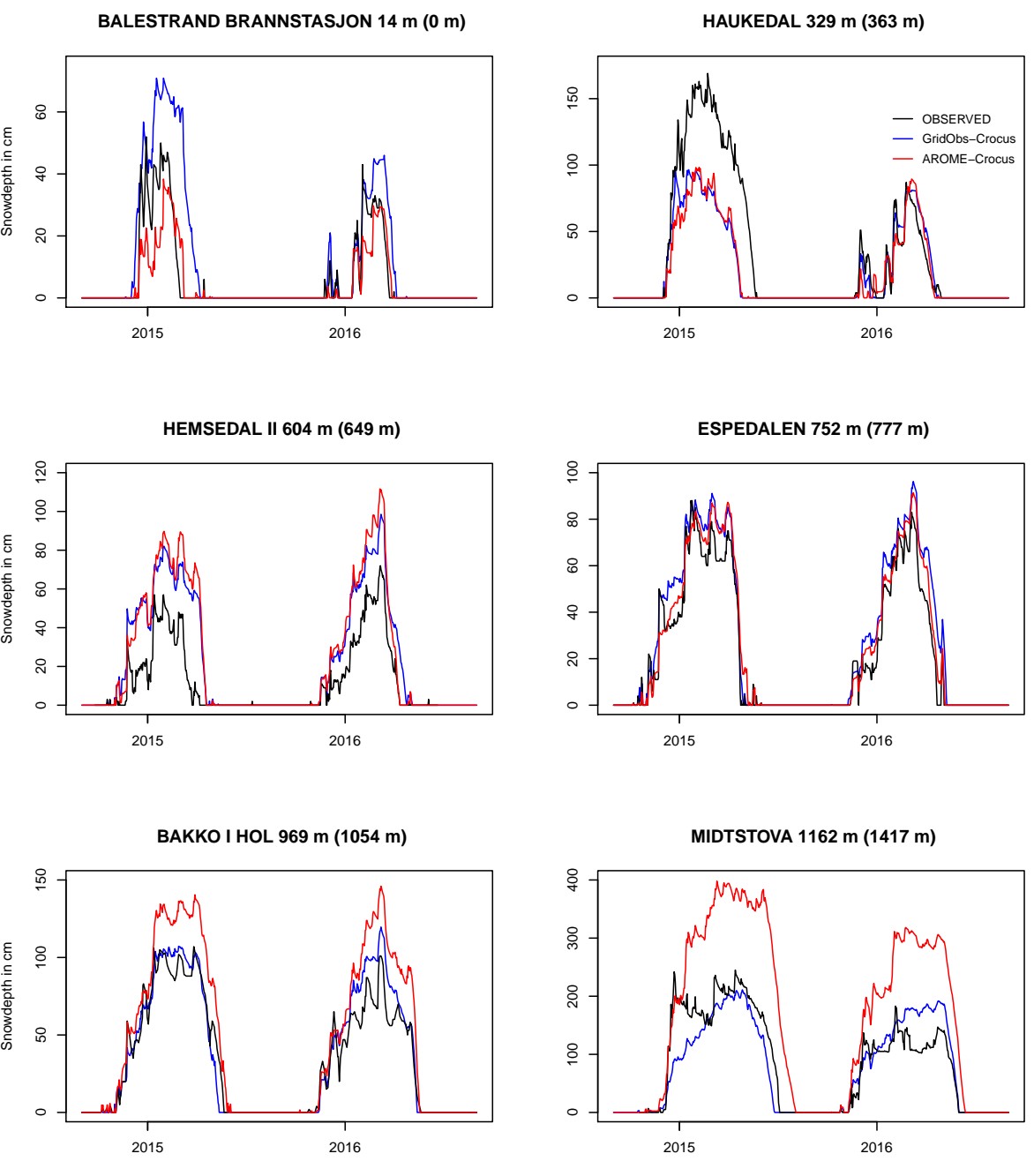

**Figure 4.** Observed and simulated snow depth (cm) at the location of six weather stations during the two winter seasons 2014-2016 (01 September 2014 - 31 August 2016) : 1) GridObs-Crocus (blue); 2) AROME-Crocus (red) and 3) observations (black). The elevation (in masl.) of the station is indicated above each plot, with in parentheses the elevation of the grid point in SURFEX/Crocus. The location of the six stations within the domain is indicated by blue dots in Fig. 1.

| | 2014-2015 | | | 2015-2016 | | |
|---|---|---|---|---|---|---|
| | Observed | GridObs Bias | AROME Bias | Observed | GridObs Bias | AROME Bias |
| Snow depth (cm) | - | +4 | +23 | - | +9 | +17 |
| Length snow season (days) | 151 | +11 | +17 | 137 | +8 | -2 |
| Date start of snow season (days) | 15 November | -2 | -2 | 15 November | +2 | +12 |
| Date end of snow season (days) | 02 May | -3 | +7 | 25 April | +2 | -3 |
| Date max snow (days) | 30 January | +13 | +17 | 22 February | +13 | +7 |
| Max snow (cm) | 112 | 0 | +17 | 88 | +9 | +18 |

**Table 2.** Bias in snow depth, length of snow season (defined as number of days with more than 5 cm snow depth), start of snow season (defined as first day with more than 5 cm snow), end of snow season (defined as the day after the last day with more than 5 cm snow), the date for the maximum snow depth and the maximum snow depth. The two snow seasons run from 01 September 2014 to 31 August 2016. A negative bias in days means a too early date for the start/end/max snow, and a positive bias in days means a later date compared to observations. GridObs-Crocus is abbreviated to GridObs and AROME-Crocus to AROME.

that is not part of the gridded observation dataset is interesting.

The strong overestimation at Midtstova (see Fig. 4) by AROME-Crocus can be explained by the fact that Midtstova is located in an area with systematic and relatively large overestimation of precipitation in AROME-MetCoOp. In addition, AROME-MetCoOp underestimates the temperature by about 2 degrees in this area during winter. This can be seen in verification reports of the AROME MetCoOp model, for example in Homleid and Tveter (2016). In the forcing data for Midtstova we find a bias of -1.5 degrees for AROME-Crocus, compared to -0.8 degrees for GridObs-Crocus. This bias is larger than the overall bias for all nine stations measuring temperature: -0.5 degree for AROME-Crocus and -0.2 degree for GridObs-Crocus. During the snow accumulation season the temperature at Midtstova is mostly well below freezing level. There are a few episodes each winter with temperatures just above zero, where the underestimated temperature in AROME-MetCoOp means the precipitation during those episodes comes as snow instead of rain, but these do not add up to large amounts. Midtstova is also a high-mountain station, which is very exposed to strong wind. Redistribution of snow due to wind is not captured in the SURFEX/Crocus model. GridObs-Crocus shows much more realistic results for Midtstova, although there is an underestimation during the first part of the 2014-2015 winter. From 27 October 2014 until 26 January 2015, the precipitation sensor at Midtstova was out of order, and the forcing from GridObs-Crocus for Midtstova will therefore be represented by interpolated values from surrounding stations, which might explain the underestimation.

An evaluation of the precipitation forcing data for AROME-Crocus and GridObs-Crocus for the 30 weather stations reveals that the AROME-Crocus forcing has about 40% more rain and 20% more snow compared to GridObs-Crocus. The differences are largest for the stations above 800 meter, which often receive about 50% more snow. This can clearly be seen for Bakko i Hol and Midtstova in Fig. 4.

| RMSE | 2014-2015 | | 2015-2016 | |
|---|---|---|---|---|
| | GridObs | AROME | GridObs | AROME |
| Snow depth (cm) | 29 | 62 | 27 | 49 |
| Length snow season (days) | 25 | 32 | 21 | 24 |
| Date start of snow season (days) | 10 | 13 | 5 | 23 |
| Date end of snow season (days) | 15 | 22 | 12 | 17 |
| Date max snow (days) | 31 | 34 | 24 | 16 |
| Max snow (cm) | 30 | 61 | 28 | 53 |

**Table 3.** RMSE for snow depth, length of snow season, start of snow season, end of snow season, the date for the maximum snow depth and the maximum snow depth. The two snow seasons run from 01 September 2014 to 31 August 2016. GridObs-Crocus is abbreviated to GridObs and AROME-Crocus to AROME.

Table 2 summarizes the bias over all stations for the two winter seasons (01 September 2014 - 31 August 2015 and 01 September 2015 - 31 August 2016). The bias was calculated as the mean of the differences between simulated and observed snow depth, and only for the days where there is snow present in the observations or at least one of the experiments. GridObs-Crocus shows a significantly smaller bias (4 and 9 cm) compared to AROME-Crocus (23 and 17 cm). The maximum observed
snow depth is on average 112 cm for 2014-2015 and 88 cm for 2015-2016. GridObs-Crocus shows a very small bias (0 and 9 cm respectively), while AROME-Crocus overestimates the mean maximum snow depth by 17-18 cm. Table 3 summarizes the RMSE over all stations for the two winter seasons. The RMSE values are larger for AROME-Crocus (compared to GridObs-Crocus) for nearly all variables, except for the date of maximum snow depth for 2015-2016.

As snow depth accumulates over the winter season, a missed (or under/over estimated) snow event can influence the re-
mainder of the season. It can therefore be useful to look at daily snow depth variations instead, as was also done by Quéno et al. (2016) and Schirmer and Jamieson (2015). Figure 5 shows the categorical frequency distribution of the daily change in snow depth for six accumulation categories, five decrease categories and one category centered around zero accumulation, on a logarithmic scale. The first two accumulation categories (up to 10 cm) are overestimated in both GridObs-Crocus and AROME-Crocus. The strongest observed increase category (>40 cm) as well as the strongest decrease category (< -20 cm) are
not represented in either of the SURFEX/Crocus experiments.

SURFEX/Crocus in stand-alone mode does not account for wind-induced snow redistribution, which can be a large contributor to strong decreases in snow depth. Figure 4 showed that episodes of a decrease in snow depth (not including the snow melt at the end of the season) were not always captured well by the models, and it could be that blowing snow is the cause of
this. Following Quéno et al. (2016), two diagnostics have been applied to look into this issue: blowing snow days and melting snow days. Blowing snow days are defined as days during which the wind speed (at a height of 10 m) during the past 24 hours (between 06 and 06 UTC, as this is when snow depth measurements are made) exceeds 8 m s$^{-1}$, while the snow surface

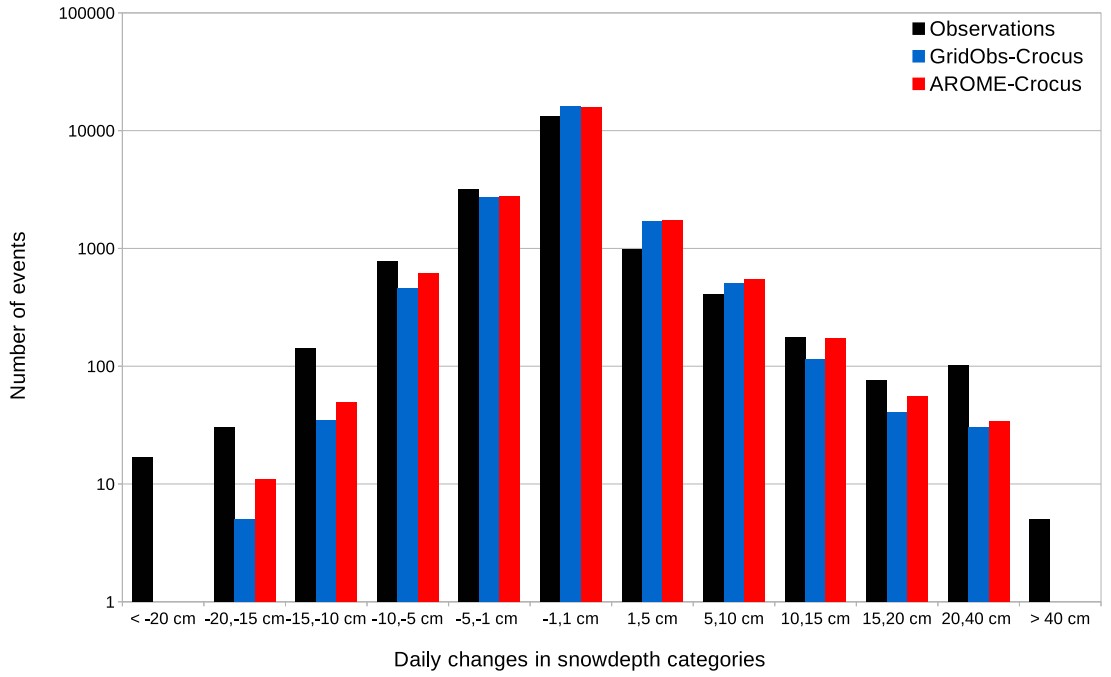

**Figure 5.** Categorical frequency distribution of daily changes in snow depth for observations (in black), GridObs-Crocus (in blue) and AROME-Crocus (in red), for all stations during 01 September 2014 - 31 August 2016. The y axis is on a logarithmic scale.

temperature is below 0 °Celsius (since only dry snow can be blown away). The temperature of the snow surface is taken from the SURFEX/Crocus output for 12 UTC each day. The wind speed is taken from AROME-MetCoOp, which is used as forcing in both SURFEX/Crocus experiments. The modeled wind speed is used because only 6 out of 30 stations used in this study observe wind speed. When comparing the forecasted maximum wind speed from AROME-MetCoOp with the observed max-
5     imum wind speed from these 6 stations, we find a slight overestimation by AROME-MetCoOp (a bias of 0.3 m/s). Blowing snow days and non-blowing snow days are correctly identified in 94% of all days, with a hit rate of 0.86 and a false alarm rate of 0.04. This shows that the modeled wind speed can be used to determine blowing snow days. The wind threshold of 8 m s$^{-1}$ for dry snow transport is taken from Li and Pomeroy (1997). Figure 6 shows the cumulated amount of the daily changes in snow depth for 5 categories of decreasing snow depth for blowing snow days and for all days where the snow depth decreases,
10     for observations and for GridObs-Crocus, as well as the percentage of snow depth loss due to blowing snow. The cumulated amount of snow decrease is underestimated for nearly all categories. For the strongest decreasing rate (more than 20 cm), the observations indicate that 51% of the decrease in snow is caused by blowing snow. This category is not represented by GridObs-Crocus. For GridObs-Crocus, blowing snow days only contribute to the smallest decrease categories. In total (over all categories), blowing snow days contribute to 17% of the cumulated decrease in snow depth in the observations, while this

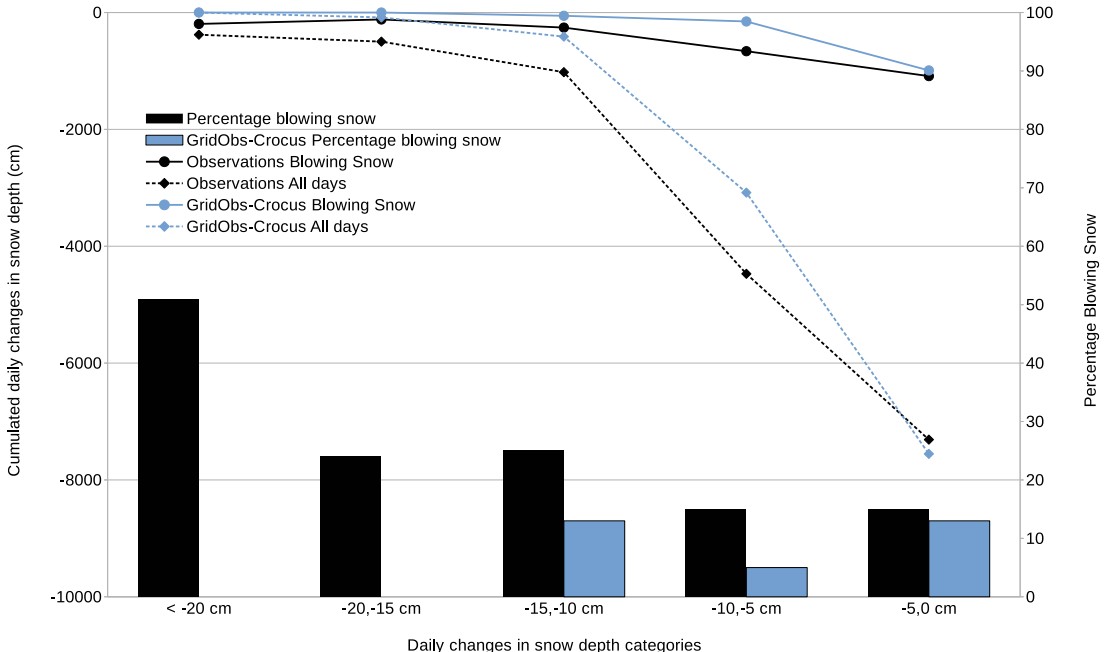

**Figure 6.** Cumulated daily change in snow depth for observations (in black), GridObs-Crocus (in blue) for all stations during 01 September 2014 - 31 August 2016, for blowing snow days (solid lines) and all days with decreasing snow depth (dashed lines). The columns show the percentage of snow loss that is caused by blowing snow, for observations (black) and GridObs-Crocus (blue).

amounts to 10% in GridObs-Crocus.

Melting snow days are defined as days when the simulated surface temperature of the snow is 0 °Celsius. Figure 7 is similar to Fig. 6, but for melting snow days. For GridObs-Crocus, melting snow is the main responsible factor contributing to
a decrease in snow depth. The largest decrease category is not represented by GridObs-Crocus, but for the other categories, melting snow is responsible for 57 - 100% of the decrease in snow depth. This is not surprising as SURFEX/Crocus does not represent blowing snow, so decrease in snow depth is caused by either snow melt or other processes such as snow compaction. The cumulated daily changes in snow depth for melting snow days as well as all days with a decrease in snow depth are underestimated by GridObs-Crocus for all categories except the smallest one (less than 5 cm loss in snow depth). This shows
there is a general underestimation of snow ablation, as well as an underestimation of snow melt in GridObs-Crocus. The same goes for AROME-Crocus (not shown).

SURFEX/Crocus does have an option to run with sublimation in case of snowdrift. This option has been tested for two stations from Fig. 4: Midtstova and Hemsedal II. In this experiment, SURFEX/Crocus was run twice in 1D mode for these 2 locations: one experiment with identical settings as AROME-Crocus, and one nearly identical with the exception of the option
for sublimation in case of snowdrift (AROME-Crocus+BS). The results are shown in Fig. 8. For both locations, the snow depth

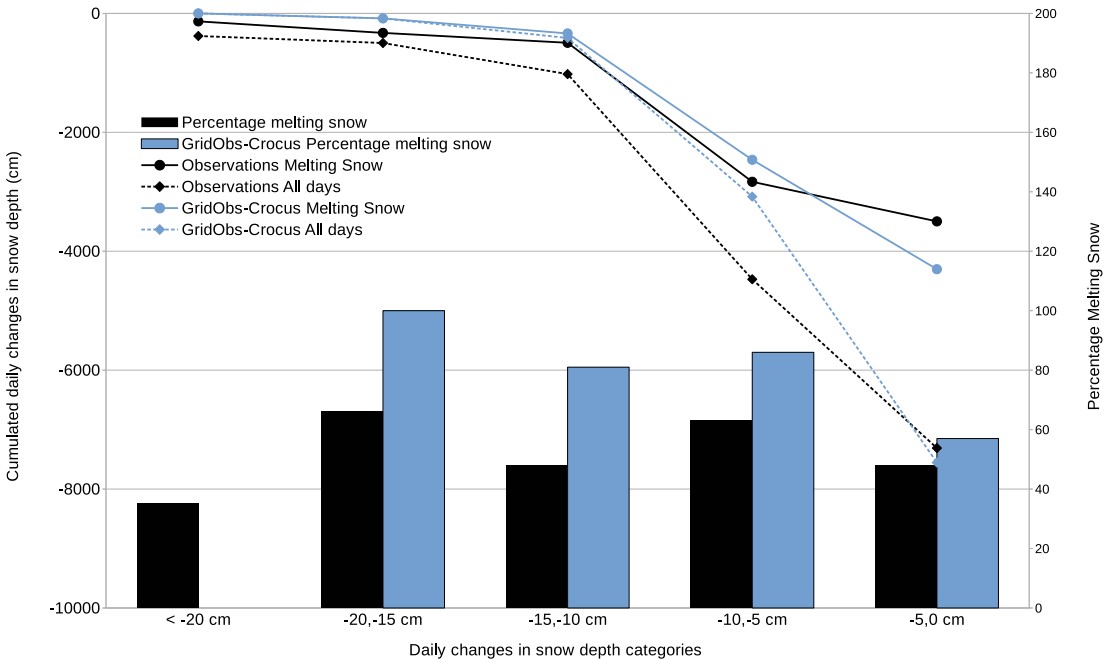

**Figure 7.** Cumulated daily changes in snow depth for observations (in black), GridObs-Crocus (in blue) for all stations during 01 September 2014 - 31 August 2016, for melting snow days (solid lines) and all days with decreasing snow depth (dashed lines). The columns show the percentage of snow loss that is caused by melting snow, for observations (black) and GridObs-Crocus (blue).

in AROME-Crocus+BS is decreased, as expected. For both locations, this is an improvement. The bias in AROME-Crocus was +13 cm for Hemsedal II, which has improved to +5 cm in AROME-Crocus+BS. For Midtstova, AROME-Crocus significantly overestimates the snow depth (bias: +104 cm), this is improved in the AROME-Crocus+BS experiment (+83 cm), although the overestimation is still very large. The length of the snow season is reduced by a few days for both stations and both years,

5   similar to the results found in Brun et al. (2013). However, for Midtstova, using blowing snow sublimation does not improve the AROME-Crocus experiment to the extent that it performs equally well as GridObs-Crocus.

### 3.2 Characteristics of the snow season

Statistics for the snow season duration are shown for the two snow seasons 2014/2015 and 2015/2016 in Table 2. The length of the snow season is defined as the number of days with more than 5 cm snow during a season. For GridObs-Crocus, the

10   length of the snow season is overestimated by 8-11 days (see table 2), while AROME-Crocus overestimates the length of the snow season by 17 days in 2014-2015 and underestimates by only 2 days in 2015-2016. The same positive bias as found for AROME-Crocus in the 2014-2015 season was found by Vionnet et al. (2016). One possible explanation of this bias is the fact that the SURFEX/Crocus model assumes a uniform snow cover from the moment snow is present on the ground, and therefore shows less variability in snow cover compared to observations. In observations, there is often a period where the snow cover

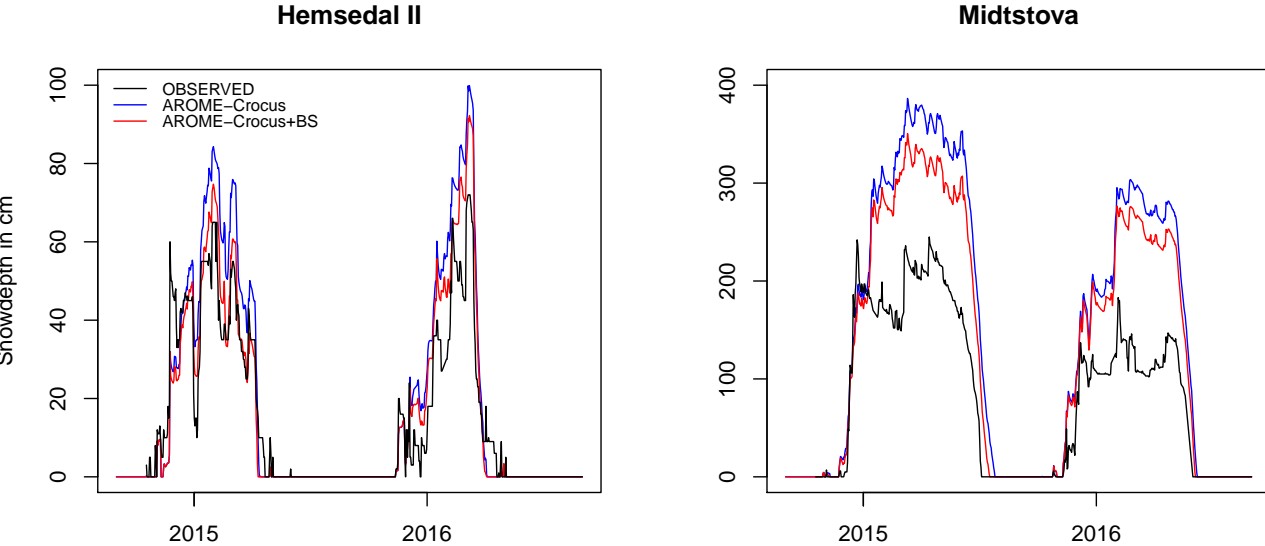

**Figure 8.** Observed and simulated snow depth (cm) at the location of Hemsedal II and Midtstova during the two winter seasons 2014-2016 (01 September 2014 - 31 August 2016) : 1) AROME-Crocus 1D experiment (blue); 2) AROME-Crocus+BS 1D experiment with sublimation loss during blowing snow events (red) and 3) observations (black).

fluctuates - for example thinning to below 5 cm after the first snow has fallen and before a continuous snow cover has been established for the winter season. AROME-Crocus predicts the length of the 2015/2016 season really well though.

The start of the snow season is defined as the first day with more than 5 cm of snow, and the end of the snow season as the day after the last day with more than 5 cm of snow. A negative bias in the start of the snow season means a too early start, while a positive bias means a too late start of the snow season. GridObs-Crocus has a bias of only two days (negative for the first winter and positive for the second winter) for the start of the snow season, while the snow season starts up to 12 days too late in AROME-Crocus during the second year (the first year has a bias of -2 days). The season ends on average in late April or early May. In GridObs-Crocus, the season ends 3 days early during the first year, and 2 days late during the second year. In AROME-Crocus, this is 7 days late and 3 days early respectively. The observed maximum snow depth occurs on average at the end of January during the first year, and late February in the second year. Both experiments show a later date for the maximum snow depth.

Figure 9 show the distribution of the bias in the start and end of the snow season, as well as the date of maximum snow depth, for all 30 stations and for two winter seasons. Most stations show a bias near zero (between -5 and +5 days) for the start of the snow season. In AROME-Crocus, the snow season sometimes starts much too late. The bias for the end of the snow

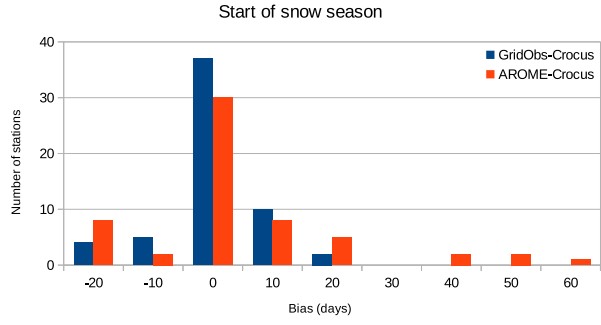

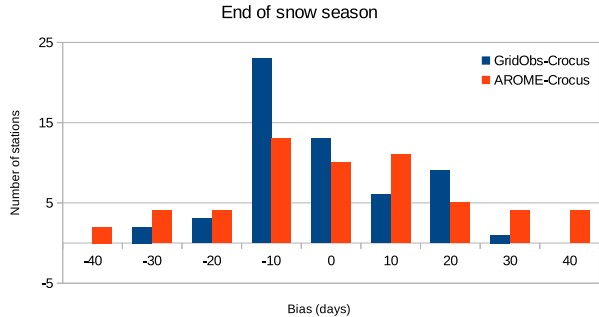

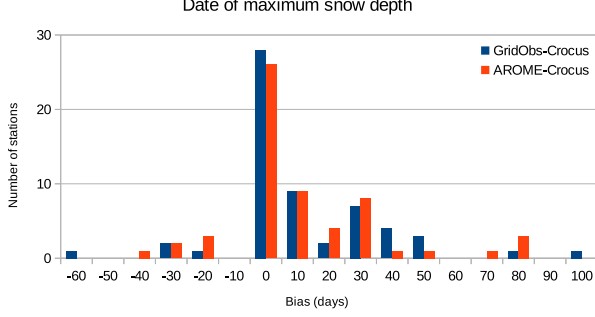

**Figure 9.** Distribution of the bias in start of snow season (top), end of snow season (middle) and date of maximum snow depth (bottom), for all 30 stations and for 2 winter seasons.

| Jaccard Index | GridObs-Crocus | AROME-Crocus |
|---|---|---|
| 15 March 2015 | 0.92 | 0.99 |
| 20 April 2015 | 0.82 | 0.93 |
| 15 May 2015 | 0.65 | 0.79 |
| 04 July 2015 | 0.19 | 0.63 |

**Table 4.** Jaccard index for the snow covered areas shown in Fig. 10. A score of 1 means the image perfectly matches the MODIS image, a score of 0 means there is no overlap between the image from the experiment compared to the MODIS image.

season shows that GridObs-Crocus often ends the snow season too early while AROME-Crocus tends to end the season too late. The bias for the date of the maximum snow depth of the season is mostly around zero for most stations and both models, but there are some outliers especially towards the strong positive bias. This is due to stations like for example Midtstova in Fig. 4, where the maximum observed snow depth occurs rather early in the season, while both experiments show a maximum much later in the season.

### 3.3 Snow-cover pattern

Figure 10 shows the spatial pattern of snow cover over the SURFEX/Crocus domain compared to MODIS data over the same area. The snow covered area is shown at different dates throughout the snow melt season: 15 March, 20 April, 15 May and 04 July 2015. On 15 March 2015, nearly the whole area is covered with snow. The only exceptions are areas right besides the fjords (white areas) in the west (well captured by both experiments), and at the bottom of valleys in the east (not captured by the SURFEX/Crocus experiments). On 20 April 2015, the snow has clearly started to melt in the valleys to the east. This is captured well by AROME-Crocus, while GridObs-Crocus shows too little snow around the valleys in the southeast of the domain. By 15 May 2015, a lot of snow had disappeared in the eastern part of the domain, while the western part has not changed much from the previous month. The average date for the end of the snow season for all the 30 weather stations for the 2014-2015 season was 02 May 2015 (see table 2), but the dates of the end of the snow season for individual stations range from 18 February (Fresvik, 32 masl.) until 05 July (Midtstova, 1162 masl.). Again, AROME-Crocus captures the snow cover pattern better than GridObs-Crocus. By 04 July 2015, the snow cover is limited to areas with higher elevation. AROME-Crocus captures the spatial pattern of snow cover very well. In GridObs-Crocus, nearly all snow has melted now, and the snow-covered area is underestimated. Earlier it was shown (in Fig. 9) that GridObs-Crocus has a negative bias (too early) for the end of the snow season for the 30 snow depth stations, while AROME-Crocus has a positive bias (too late). As discussed previously, the differences between the snowfall amounts from the two precipitation forcing datasets are largest for the highest parts of the domain, where AROME-Crocus receives about 50% more snow compared to GridObs-Crocus. This explains why the differences between GridObs-Crocus and AROME-Crocus in Fig. 10 are also largest in this area (especially by the end of the snow season on 04 July).

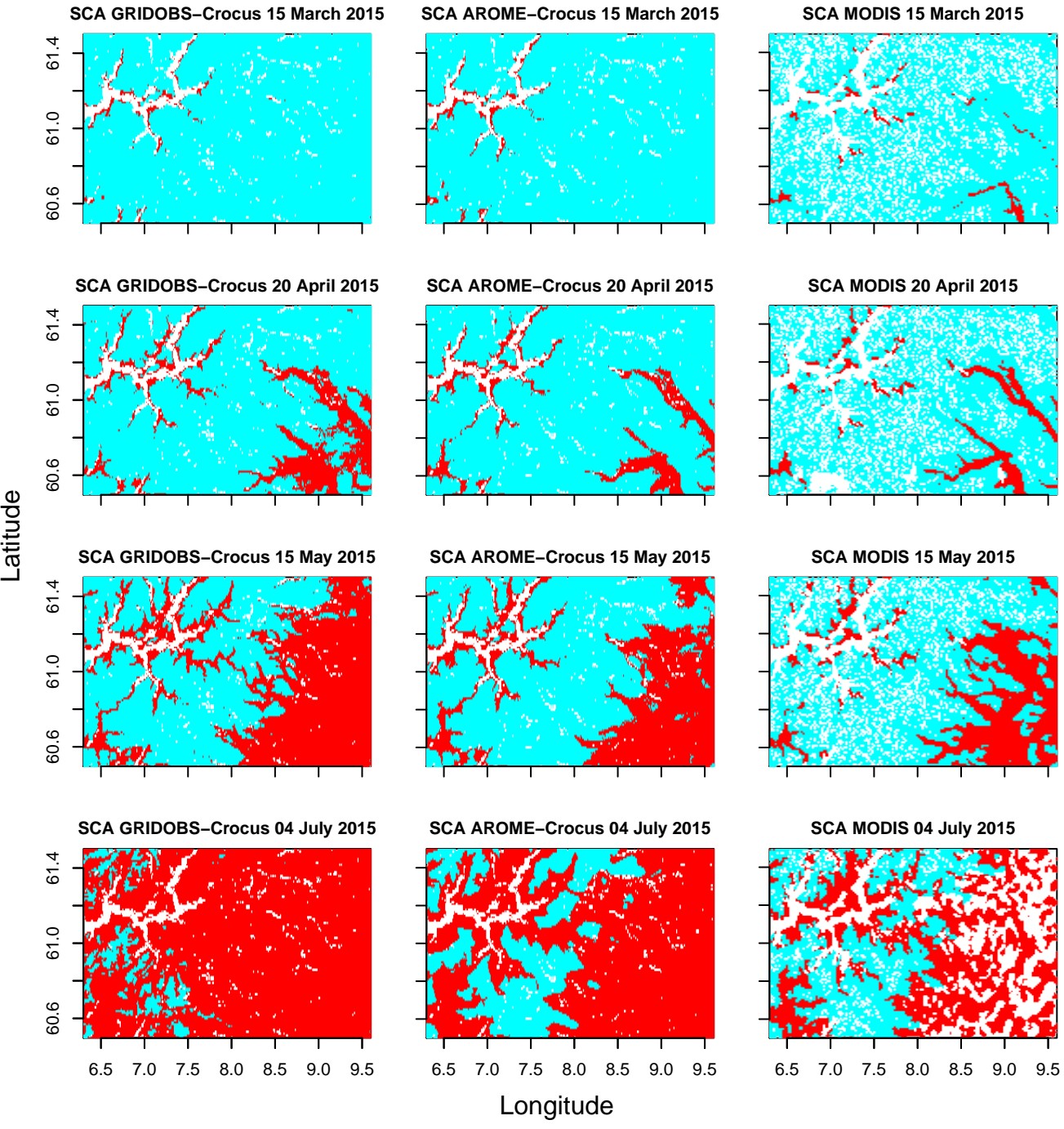

**Figure 10.** Snow covered area (where cyan is snow, red is no snow, and white is missing data or water surfaces) for GridObs-Crocus (left column), AROME-Crocus (middle column) and from MODIS satellite images (right column), for (rows from top to bottom): 15 March 2015, 20 April 2015, 15 May 2015 and 04 July 2015.

Table 4 shows the Jaccard indices for the images from Fig. 10. The Jaccard index was also used by for example Quéno et al. (2016). It is a similarity index applied to the snow cover images which were remapped onto the same grid (which means that the snow cover from the MODIS images used to calculate the Jaccard index has a lower resolution than the one shown in Fig. 10). The Jaccard index is calculated as $J(X, Y) = |X \cap Y|/|X \cup Y|$, where X and Y are the simulated and observed snow cover, respectively. The number of grid points that are snow-covered in both SURFEX/Crocus and in the MODIS image is divided by the total amount of snow-covered grid points (in either SURFEX/Crocus or MODIS). When the Jaccard index equals 1, there is a perfect match between snow-covered grid points, and when the Jaccard index equals 0, there is no match at all. Table 4 shows that AROME-Crocus consistently has higher Jaccard indices compared to GridObs-Crocus. The indices decrease (for both experiments) during the melt season.

## 4   Discussion

Although both experiments are capable of simulating the snow pack over the two winter seasons, the two simulations provide different results regarding the snow depth and the spatial snow-covered area. There is an overestimation of snow depth in the AROME-Crocus experiment, even though the snow-covered area throughout the melt season is better represented by this experiment. When using gridded observations (GridObs-Crocus), the simulation of snow depth is significantly improved, while the spatial distribution of the snow cover is highly underestimated, particularly late in the snow-melt season. There is an underestimation of snow ablation in both experiments, which is due to a combination of the absence of wind-induced erosion of snow and underestimation of snow melt in SURFEX/Crocus, and biases in the forcing data. Possible causes for these different results are further discussed below, by focusing on the quality of the model validation, the forcing data set and the snowpack model.

### 4.1   Quality of the model validation

The model validation was carried out using both snow measurements at individual weather stations and MODIS satellite images. Using several data sources to validate simulations is important as these two sources supplement each other. Stations give point validations with daily time series, while the satellite images provide images of the snow cover for an entire area for cloud free days. Even though the GridObs-Crocus simulation provides reasonable results at individual stations, the MODIS images show that the snow cover disappears too fast, particularly late in the snow melt season. This may indicate that the gridded interpolated observations of temperature and precipitation (the forcing data) are not representing the terrain variability in the study area sufficiently well. The western region is dominated by terrain with steep gradients, which requires a higher density of weather stations representing the full range of terrain elevations compared to smooth landscape areas. As described in Section 2.2, there is a low elevation bias in the national observational network with too few stations in areas above 900 masl. This increases the uncertainty of the precipitation and temperature estimates in the mountainous regions. The quality of the gridded data are obviously highest for locations closest to the stations, providing better results in those areas. An underestimation of the snow depth at high elevations would explain the underestimated snow cover during the melt season.

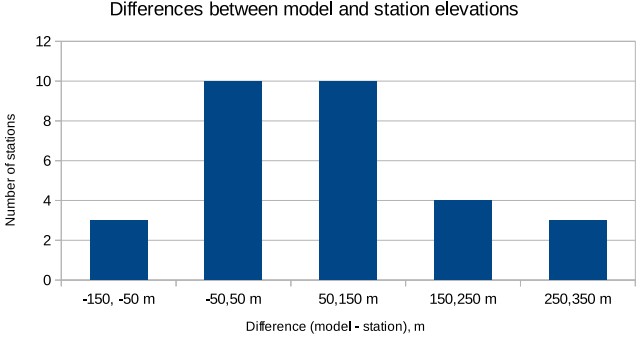

**Figure 11.** Differences between station elevation and the height of the station in the SURFEX/Crocus model.

Nearly all (29) of the 30 stations that measure snow depth also measure precipitation (7 measure hourly precipitation, while 22 measure daily precipitation), which means the observed precipitation from these stations are used in the gridded observations dataset used to force GridObs-Crocus. Only 9 out of the 30 stations also measure temperature. Although precipitation is not directly related to snow depth, and temperature also plays an important role, it could still be argued that the GridObs-Crocus results are best in the locations of the observations that are included in the gridded dataset used to force SURFEX/Crocus. The only station that was not part of the gridded precipitation dataset is Hemsedal II. The bias for GridObs-Crocus at Hemsedal II is 25 cm, which is larger than the overall bias for all stations (7 cm), but the RMSE is about the same (27 cm for Hemsedal II and 28 cm for all stations). Although this shows that the performance of a station not included in the gridded precipitation dataset is about the same as the performance of stations that are part of this dataset, one station is not enough to draw conclusions about the entire domain.

The representativity of a station location is sensitive to the terrain variability. The orography used in the SURFEX/Crocus experiments has a resolution of 1 km, leading to differences between the actual station height and the average height used for the center of the 1 km grid cell in SURFEX/Crocus. Figure 11 shows the distribution of those differences for all 30 stations. The average bias is 79 m. Most stations are placed at higher elevations in the model as compared to their actual elevation, but it should also be kept in mind that a grid point in SURFEX/Crocus describes a larger area (and range of elevation) compared to the actual observations. Especially in the mountainous region in the west of the domain (see Fig. 1), with high mountains, steep slopes and deep valleys, there may be large differences in height within a distance range of 1 km. In Section 4.2 the sensitivity of terrain effects on precipitation phase computations is discussed.

### 4.2 Quality of the forcing data sets

Raleigh et al. (2015) showed that snow simulations are more sensitive to biases in forcing data than random errors, and that precipitation bias is the most important factor. There is a negative bias in the gridded precipitation used in GridObs-Crocus (Lussana et al., 2018a), especially for data-sparse areas (e.g. high-mountainous areas) and for intense precipitation. Missing

episodes of intense snowfall would explain part of the underestimation of the snow depth in GridObs-Crocus. There are plans to improve the gridded observations of precipitation by adjusting the solid precipitation to account for the wind undercatch, and by post-processing of the predicted precipitation fields to adjust for bias (Lussana et al., 2018a). Forecasts from the AROME-MetCoOp model are known to overestimate the occurrence of precipitation events of less than 10 mm (Müller et al., 2017), and Fig. 5 showed that AROME-Crocus overestimated daily changes in snow depth up to 15 cm. An evaluation of the accumulated snowfall from the two forcing datasets showed that snowfall amounts in AROME-Crocus are about 20% higher compared to GridObs-Crocus, and even more ( 50%) for stations at altitudes above 800 masl.

To test the sensitivity of terrain effects on precipitation phase and on the snow depth simulations in regions with steep topography, we compared two methods for determining precipitation phase from the AROME-MetCoOp forecasts. First, precipitation phase was determined as already presented in our AROME-Crocus experiment (by using the post-processed air temperature as a threshold temperature, see Table 1), and second, using the raw AROME-MetCoOp snowfall and rainfall forecasts computed by the atmospheric model's own microphysics. These 2.5 km spatial resolution forecasts were then interpolated to 1 km using standard bilinear interpolation, also described in Section 2.3.1. The comparison revealed that high-resolution terrain data improved the results by reducing the amount of snow, particularly in low-altitude areas near steep terrain gradients. The bias in snow depth improved from +42 cm (using raw AROME-MetCoOp precipitation) to +20 cm (using the terrain-adjusted AROME-MetCoOp precipitation), and the RMSE improved from 68 cm (raw) to 56 cm (terrain-adjusted). Figure 12 shows the fraction of snowfall compared to the total precipitation for both methods, for the first winter season. It is clear that using the terrain-adjusted AROME precipitation results in a better representation of the terrain in the domain, as many features are lost in the coarser resolution raw AROME precipitation. Figure 13 illustrates the differences in the computed snowfall from the two methods as a function of elevation and longitude. The west-east transect from steep terrain (including fjords and mountains) in the west to smoother terrain in the east is clearly illustrated, with largest differences in the western part of the domain. In these areas a more realistic precipitation phase (rainfall) was more frequently computed than when using the raw AROME-MetCoOp precipitation phase. This emphasizes the importance of terrain-adjusting the forcing data from NWP models for obtaining more correct precipitation phases.

Precipitation phase in both AROME-Crocus and GridObs-Crocus was determined using a fixed threshold temperature of 0.5 °Celsius to distinguish between rainfall and snowfall. This simplification represents an uncertainty which could result in some actual snow events characterized as rainfall, and to a lesser extent the other way around. Future studies could focus on studying alternative ways to determine the precipitation phase by better exploiting the microphysics of the NWP model. Generally, our experiments show that highest potential for obtaining good snow simulations for larger regions lies in improving the forcing data, and particularly by improving the raw NWP data by post-processing techniques. The use of gridded interpolated observations alone, as was tested in GridObs-Crocus, displays some limits for snow-cover mapping over larger regions. There is a large potential to improve NWP forcing by combining different data sources, e.g. assimilation of various observations (weather station data, precipitation radars etc.).

Sauter and Obleitner (2015) investigated the sensitivity of SURFEX/Crocus snowpack modeling on Svalbard (Arctic Norway) to input parameters, and found that for higher elevations (in the accumulation zone), precipitation and radiation are the

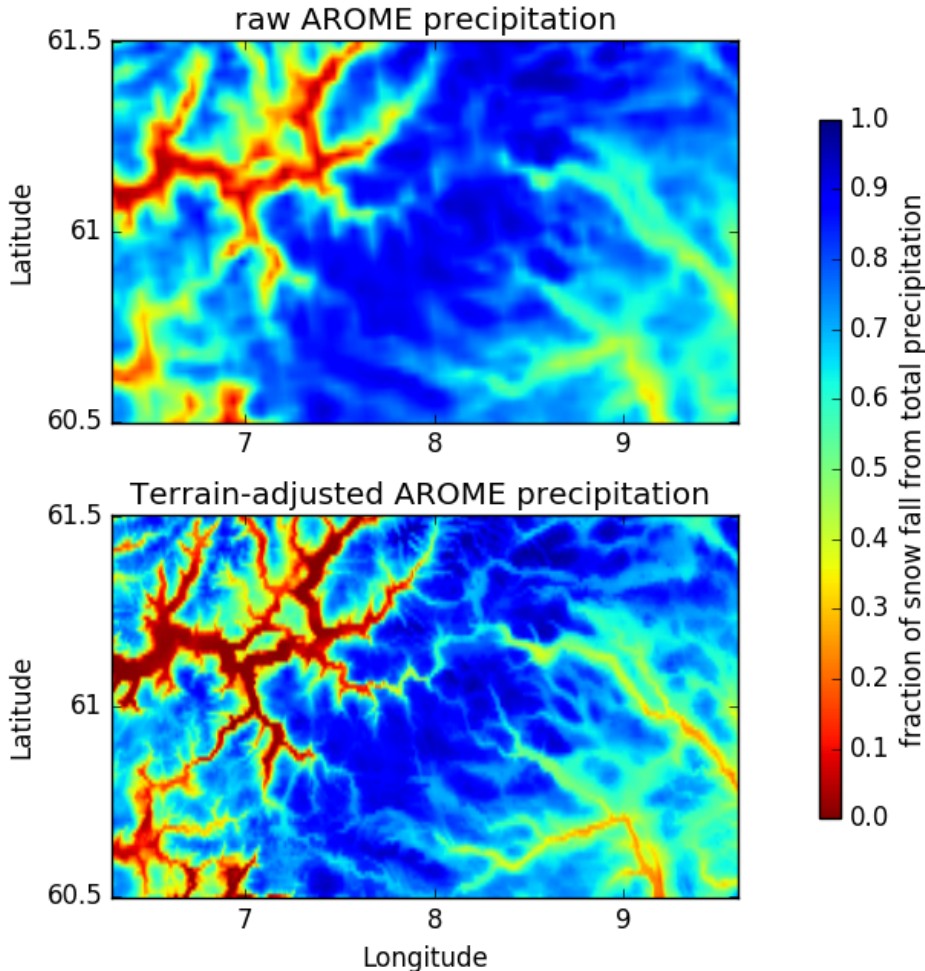

**Figure 12.** The fraction of accumulated snowfall from total accumulated precipitation computed for the period 01 September 2014 to 31 May 2015 . Rainfall and snowfall are computed using: 1. Raw AROME-MetCoOp snowfall and rainfall forecasts at 2.5 km spatial resolution interpolated to 1 km (top); and 2. a threshold temperature of +0.5 °Celsius, using the temperature from the 1 km post-processed AROME-MetCoOp temperature (the terrain adjusted AROME-Crocus experiment presented in Table 1, bottom).

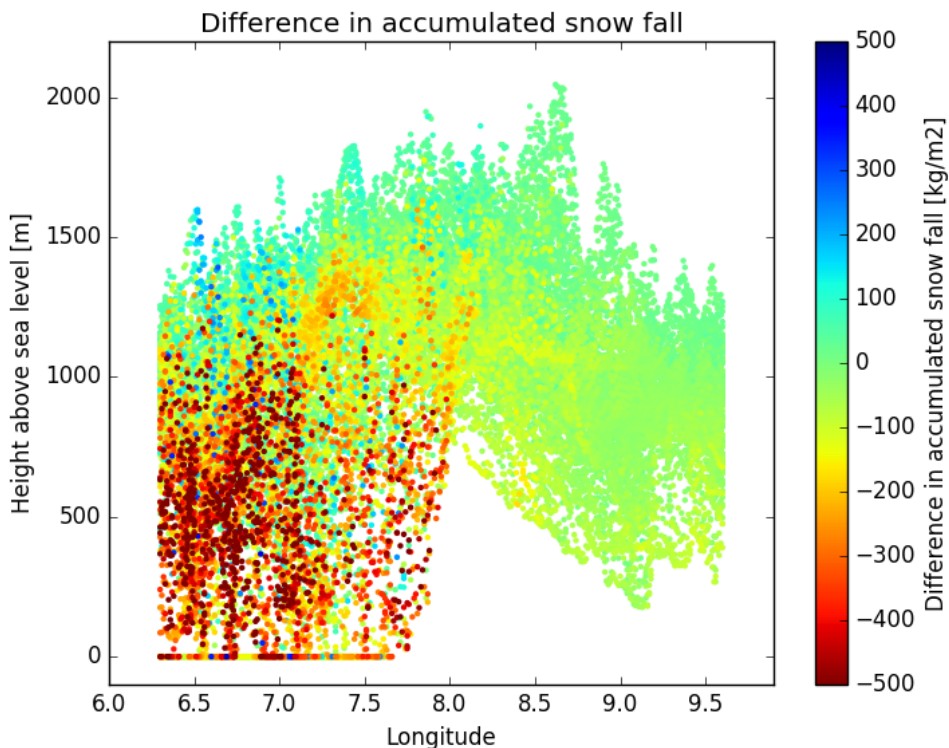

**Figure 13.** Differences in computed snowfall from two different methods as a function of elevation and longitude. The precipitation phase determination methods are the same as described in Figure 12. Negative (red) values indicate less snowfall and thus increased rainfall with the terrain-adjusted precipitation phase determination, positive (blue) values indicate more snowfall (decreased rainfall).

key factors in the evolution of the snowpack and contribute most to the model uncertainty. At lower elevations, precipitation was less important but factors such as wind speed or surface roughness increased in importance. Quéno et al. (2017) used satellite products of incoming solar and longwave radiation to force the SURFEX/Crocus model, however they concluded that improved meteorological forcing does not always lead to more accurate snowpack simulations, due to error compensations

5 within the atmospheric forcing and the snowpack model.

### 4.3 Quality of the snowpack model

The SURFEX/Crocus model assumes a uniform snow cover when SWE reaches the relatively low threshold of 1 kg m$^{-2}$. The SURFEX/Crocus model was originally developed for use in high alpine regions, where there is not a lot of vegetation. In those areas, the assumption of the uniform snow cover is realistic, as there is no interaction with vegetation, but for areas covered with

10 forest and closer to sea level this could lead to an overestimation of the snow cover. When the snow cover is overestimated, the

albedo will be too high and this will slow down the snow melt at the end of the season. This might explain the underestimated snow melt in both experiments.

The SURFEX/Crocus model grid is a collection of independent grid points with no transport of snow or other variables between grid points. It is therefore not possible to simulate the redistribution of snow by wind. It can be argued that with a resolution of approximately 1 km, the drifting snow would anyway be redistributed within the area of a grid point and not transported to neighboring grid points. Vionnet et al. (2014) showed that for explicit simulation of wind-induced snow transport a spatial resolution of less than 50 meters is required. This is currently not a feasible option for snowpack simulations over larger domains. There is an option in the SURFEX/Crocus model to calculate the rate of sublimation in case of snowdrift, which results in a loss of snow. This option was tested for two stations in his study, using the AROME-Crocus forcing dataset. As expected, this resulted in a decrease in snow depth and a decrease in season length. This is an improvement as it reduces the overestimation by AROME-Crocus, but only to some degree.

Figure 5 showed that both SURFEX/Crocus experiments underestimate the melting of snow, further supplemented by Fig. 7 for the GridObs-Crocus experiment. Underestimated melting was also found by Quéno et al. (2016, 2017) and Vionnet et al. (2016), and complementary studies are needed to investigate the cause of this issue.

Lafaysse et al. (2017) developed an ensemble snowpack model using SURFEX/Crocus called ESCROC (Ensemble System Crocus) to address modeling errors. They found that by using optimal members they were able to explain more than half of the simulation errors, and those ensembles have a significantly better predictive power than the classical deterministic approach. For future work, it would be interesting to use ESCROC and investigate the effect of different physical settings of SURFEX/Crocus. In addition, since November 2016, AROME-MetCoOp is run as an ensemble with 10 members, called MEPS (MetCoOp Ensemble Prediction System). This means that an ensemble of meteorological forcing is another possible direction for future work. Vernay et al. (2015a) used the 35 members of the ensemble prediction system based on the French NWP model ARPEGE as forcing to the SURFEX/Crocus model. The results indicated that accounting for the uncertainty in meteorological forecast significantly improves the skill and the usefulness of the model chain.

## 5 Conclusions

In this study we have evaluated the performance of the SURFEX/Crocus snow model for a region in South Norway covering both steep terrain gradients with fjords and high-mountain areas in the western parts as well as smoother terrain in the eastern parts. The experiments tested different types and combinations of forcing data (raw numerical weather predictions, post-processed weather predictions and gridded observations): 1. AROME-Crocus, which used weather forecasts from the AROME-MetCoOp model, including post-processed air temperature and wind speed, and 2. GridObs-Crocus, which used gridded observations of temperature and precipitation combined with meteorological forecasts from AROME-MetCoOp. Snow simulations were carried out for two years (01 September 2014 - 31 August 2016). The main findings are as follows:

- GridObs-Crocus provides the best estimates of the snow depth at individual stations with bias of 7 cm and RMSE of 28 cm. AROME-Crocus has a bias of 20 cm and RMSE of 56 cm.

- AROME-Crocus provides the best representation of the spatial distribution of snow cover, particularly during the melting season. In GridObs-Crocus the spatial snow cover distribution is captured during winter, but underestimation of snow depth at high elevations (due to the low elevation bias in the gridded observation dataset) is likely causing the snow cover to decrease too soon during the melt season, leading to unrealistically little snow by the end of the season.

- Forcing data consisting of post-processed NWP data (observations assimilated into the raw NWP weather predictions) are most promising for snow simulations, when larger regions are evaluated. Post-processed NWP data (AROME-Crocus) provide a more representative spatial representation for both high mountains and lowlands compared to interpolated observations (GridObs-Crocus).

- In regions with steep terrain gradients, terrain-adjustment of precipitation phase is highly important for improving the rainfall and snowfall determination when using NWP data.

- Blowing snow (which is not simulated by SURFEX/Crocus) contributes to 17% of all decreases in snow depth, and to 50% of the strongest decreases of more than 20 cm of snow depth loss in a day. Using the option in SURFEX/Crocus of running with sublimation in case of snowdrift is not enough to address this issue.

To investigate the impact of using gridded observations of temperature and precipitation separately, "leave-one-out" experiments could be carried out (two extra experiments where one uses only gridded observations of temperature, and one uses only gridded observations of precipitation, while all other variables come from AROME-MetCoOp). Using the multi-physical ensemble system ESCROC (Ensemble System Crocus), and/or an ensemble of meteorological forcing would be an another interesting topic for future work. Finally, when using AROME-MetCoOp as forcing data for running SURFEX/Crocus at a resolution higher than 2.5 km, terrain adjustment routines should be applied to the generation of forcing data. In this study we accounted for local terrain effects, by using post-processed AROME-MetCoOp temperature and wind, but this could be extended to other variables.

The findings in this study have improved our understanding of regional snow modeling in Norway, which is important for not only water resource planning and flood forecasting, but also for impact studies related to climate change and winter climate. Running the SURFEX/Crocus model in gridded version for Norwegian conditions using a combination of data sources (raw and post-processed weather predictions and observations) is very promising. The result from this study is very valuable information which may be used for future development of a system for daily snow mapping in Norway.

*Data availability.* Snow depth and meteorological variables from the stations used in this study are freely available through https://frost. met.no/ (Frost, 2018). AROME-MetCoOp forecasts are available through http://thredds.met.no/thredds/metno.html. The gridded dataset of temperature and precipitation is available at http://thredds.met.no/thredds/catalog/metusers/senorge2/seNorge2/archive/catalog.html. Hourly temperature and precipitation data is available from 2010 up to the present day. For daily temperature and precipitation data, the archive goes back to 1957 and can be downloaded at http://doi.org/10.5281/zenodo.845733. The data are also shown on the web-portals www.senorge.no

and www.xgeo.no (both in Norwegian only). The SURFEX-Crocus simulations for both experiments can be made available for research purposes by contacting the authors.

*Competing interests.* The authors declare that they have no conflict of interest.

*Acknowledgements.* The authors are grateful for the funding of this study by the Research Council of Norway through the project "Better
5  SNOW models for prediction of natural hazards and HydropOWer applications" (SNOWHOW), led by Thomas Skaugen at the Norwegian Water Resources and Energy Directorate (NVE). We would like to thank our fellow participants in the SNOWHOW project for invaluable help and discussions: Thomas Skaugen, Tuomo Saloranta (NVE), Karsten Müller (NVE), Kjetil Melvold (NVE) and Sjur Kolberg (SINTEF). In addition, we would like to thank Tuomo Saloranta for providing the processed MODIS images, and Cristian Lussana (MET Norway) for valuable help and discussions. We are also very grateful to two anonymous reviewers, for their time, effort and very helpful suggestions,
10  which resulted in a greatly improved paper.

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
