# Peer review of "Forcing the SURFEX/Crocus snow model with combined hourly meteorological forecasts and gridded observations in southern Norway"

_The Cryosphere, 2017_

## Referee Comment (RC1) · Anonymous Referee #1 · 30 Nov 2017

This article, entitled "Forcing the SURFACE/Crocus snow model with continued hourly meteorological forecasts and gridded observations in southern Norway", has for its main objective to improve numerical prediction of terrestrial snow in Norway. The approach that is presented and examined is based on the used of an external land surface system, SURFEX, that includes the sophisticated snow model CROCUS, and which is driven by a combination of surface analyses (for precipitation and air temperature) and of short-range numerical atmospheric prediction.

This subject is certainly of interest. Better analysis and prediction of terrestrial snow

remains one of the great challenges that national environmental prediction centers face. But unfortunately the scope and ambitions of this study are insufficient, in the sense that this kind of work has often been presented in previous papers (which are not cited), and are actually used Operations in several centers. While the results analysis is very well done, with interesting metrics and with interesting discussion, the overall impact (positive) is not Earth shattering.

Unless the authors come up with a major overhaul of the article, maybe based on some of the more specific comments below, this paper should be rejected.

Specific comments: —————

Page 2, line 25: The main objective of the paper is stated as: "The aim was to compare and combine different forcing data sets as input to the SURFEX/Crocus model and validate the computed accumulated snow amounts and snow melt patter in both Norwegian mountains and lowlands." The authors have to realize that this type of work has been done often. Practically every snow scheme that has been developed in the last few decades have been tested at observational surface stations using atmospheric forcing that are not unlike what is used in this study. Many land data assimilation systems also use snow models forced with a mixture of observations and model predictions to produce terrestrial snow analyses. In order to make this article acceptable for publication, the authors need to describe and include these previous studies in their Introduction, and explain how exactly what they have done contributes in an original manner to advancing this body of research.

Page 4, Table 1: Was there any adaptation applied to the atmospheric forcing, e.g., for air temperature, or precipitation phase? My understanding that the atmospheric model that provided some of the forcings was at lower horizontal resolution than the external land surface model. There might be some inconsistencies in terrain height (between model and reality) that could lead to biases.

Page 7, line 8: "... is in good agreement with the observations, ..." I don't know how

we could say that from Fig. 3. More generally, more discussion is needed to correctly present and understand what is shown in that figure. For instance, it seems there is a significant bias for snow depth below 100 cm. Also, it might be better to show plot the density of points rather than just the cloud of points.

Page 9, line 9: "... matches the observed pattern of increases and decreases more closely than AROME-Crocus." I disagree with that statement... or I would say instead that this improvement is quite marginal. Is it what we should expect in terms of impact for points that do not benefit of having a surface observational station?

Page 16, line 1: "The results are promising." Too vague.

Page 16, line 1: "Both experiments are capable of simulating the snow pack over the two winter seasons." Based on current and recent scientific and technological achievements in this research area, should we consider this an achievement worth being presented as a conclusion?

Page 16, line 16: "... it could still be argued that the GridObs-Crocus are best in locations of the observations that are included in the gridded dataset used to force SURFACE/Crocus." This statement by the authors is not substantiated by evidence in this paper.

Page 17, line 28: "... this could lead to an overestimation of the snow cover. This in turn would lead to an overestimation of the snow depth." How does that work? What's the physical link here, to explain this cause and effect?

General comment: The impact of precipitation and air temperature observations on the simulation could be better highlighted with "leave-one-out" experiments.

General comment: The article would gain in quality if a comparison of the results presented in this article would be compared with what is currently available (operationally) in Norway.

General comment: Has there been any tests to evaluate the impact of air temperature

and precipitation seperately?

Minor comments: —————————

Page 1, line 8: "The results are promising." This is too vague.

Page 2, line 8: "... include more statistics to capture the physical snow processes". The word "capture" is not appropriate in that sentence.

Page 2, line 10: "grid spacing" instead of "resolution".

Page 2, line 11: "levels" instead of "layers".

Page 2, line 14: "... the atmospheric part..." this is too vague.

Page 7, Fig. 3: The text on this figure is too small, unreadable.

Page 10, Fig. 6: The two colors chosen for this figure are too alike (difficult to distinguish for old eyes like mine...)

Page 11, Table 2: The statistics presented in that table are quite interesting, but reading it is a bit tedious. I wonder if there could be another way of arranging the table.

Page 11, line 3: "... exceeds 8 m/s..." this is for the winds at what height, 10 m? This should be mentioned.

Page 12, Fig. 7: same comment as Fig. 6 concerning the colors.

Page 15, Fig. 9: the text is too small, unreadable.

---

## Referee Comment (RC2) · Anonymous Referee #2 · 18 Dec 2017

This paper presents an evaluation of the ability of the snowpack model Crocus to simulate snow depth and snow cover in southern Norway for two winters using different atmospheric driving data: (i) short-range high resolution weather forecast generated by the AROME MetCoOP system and (ii) gridded datasets for precipitation and temperature derived from observations. The authors propose an evaluation of model results using snow depth observations collected at 30 stations across the simulation domain and MODIS snow cover data at 500-m grid spacing. Daily snow depth variations are considered as in Quéno et al. (2016) to discuss more in details the physical processes

responsible for differences between simulated and observed snow depth.

The subject of this paper is interesting for the snow and mountain hydrology community because of the growing use of high-resolution weather forecast to drive snowpack models in mountainous terrain (e. g. Bellaire et al., 2011, 2013; Bernier et al., 2012; Carrera et al., 2009; Horton and Jamieson, 2016; Quéno et al, 2016; Vionnet et al, 2016). The analysis of results presented here is similar to the studies by Quéno et al (2016) and Vionnet et al. (2016) and reveals consistent and interesting model behavior between the French and the Norwegian mountains. My main comments about this study concern (i) the comparison between the different atmospheric driving datasets, (ii) the interpolation of AROME forecast on the 1-km grid used for Crocus simulations, (iii) the selection of stations for model evaluation and its impact on the analysis of model results and (iv) the originality of this work compared to other studies using AROME and Crocus in the French mountains. These questions need to be clarified prior to publication in TC. They are listed below as general comments followed by more specific and technical comments.

General comments

The comparison between simulated snow depth and snow cover using different precipitation and temperature forcing is interesting and illustrate well the strong impact of these variables on simulated snowpack evolution. However, the authors only present the results of snowpack simulations and never compare for example the precipitation forcing in their two experiments; how they differ for different elevation ranges or distance to the sea. Such comparison would be really useful to better understand the differences obtained in the simulated snowpack evolution. For example, Figure 9 shows that the snow cover remains longer at high-elevation in AROME-Crocus compared to GridObs-Crocus. Is it explained by lower precipitation at high-altitude in the GridObs forcing compared to the AROME forcing? This is not mentioned in Section 3.3 and in the conclusion and should be added to the paper.

[Figure]

AROME forecast at 2.5 km are interpolated bilinearly on the 1-km grid used for Crocus simulation. No downscaling is performed to account for the differences between the interpolated terrain height from the 2.5-km grid and the actual terrain height in the 1-km grid. This can potentially lead to large errors in region of complex terrain. For example, the phase partitioning simulated by the AROME cloud microphysical scheme is only valid at the elevation of the grid cell on the 2.5 km grid. A first order correction using a simple lapse rate is required to adjust the phase if the elevation difference is large. Overall, I recommend the author to include simple terrain adjustment routines in their AROME-Crocus simulation to use a meteorological forcing valid on the 1-km grid where are performed the snowpack simulation. This can be done using very simple methods such as the ones used in Bernier et al. (2011).

Section 4.1 shows that the authors include in their analysis stations with large differences between the model and the actual height at station location (up to 450 m). They did not make a selection of stations based on a maximal value for the difference between the model and the actual height. In their studies, Vionnet et al (2016) and Quéno et al (2016) used for example a maximal elevation difference of 150 m in absolute value. In this paper, 13 stations among 30 correspond to this criteria. What is the impact of these large elevation differences on the evaluation of model results? For example, for the stations with an elevation difference above 250 m, what is the impact on the evaluation of snow cover duration? What about the wind speed simulated at these stations and used to determine the occurrence of blowing snow days? As mentioned in my previous comment AROME forcing are only interpolated bilinearly on the 1-km grid. Therefore, altitude differences between the station height and the elevation in the interpolated terrain at 1 km from the 2.5-km grid can be potentially even larger. The authors only mention the elevation differences in the discussion (Section 4.1). I think this should be mentioned earlier in the paper; for example in Section 2.3.1 when presenting the snow depth observations. Overall, the effects of these large elevation differences should be better quantified.

The simulation framework and evaluation methods presented in this paper are very similar to the ones used by Vionnet et al. (2016) and Quéno et al. (2016) who used AROME to drive Crocus snowpack simulations in the French Alps and the Pyrenees. It is interesting to see that similar results are obtained in a different mountainous environment. However, the author need to better insist on the originality of their study compared to these previous work. It would have been interesting to see additional experiments. For example the authors used a succession of forecast from +3 to +8 to drive Crocus. Vionnet et al. (2016) and Queno et al (2016) combined daily forecast from +6 to +29 issued at 00 UTC to drive Crocus. The impact of these choices on model results has never been discussed and it would be an interesting contribution. Similarly, the authors mentioned at the end of their paper (P 19 L7-8) the potential importance of blowing snow sublimation for the high-altitude part of their domain. I recommend the authors to carry out an experiment where they test the impact of the parametrization of sublimation loss during blowing snow events implemented in Crocus. The authors could discuss the impact in terms of snow cover duration and compare it with MODIS images. Overall, these additional experiments would bring interesting insights and strengthen the discussion section which is so far very similar to the discussions in Vionnet et al. (2016) and Quéno et al. (2016).

Specific comments

Abstract: The abstract is rather vague and should present some precise figures such as the overall snow depth bias for the two experiments and the number of stations used for model evaluation. A L11-13, the authors mention the assimilation of snow depth data directly into Crocus. This topic is mentioned here but never discussed in the paper. If the authors want to keep this sentence in the abstract, they need at least to discuss more the assimilation of punctual snow depth data in distributed snowpack simulations in the discussion part.

Introduction: The current introduction of the paper does not described well enough the context of the study and the scientific questions the authors are investigating. For

example, the authors never mention the growing use of high resolution NWP forecast to drive detailed snowpack model in mountainous terrain and the limitations associated with these systems. In particular, previous studies using AROME forecasts to drive Crocus in the French Alps and the Pyrenees (Quéno et al., 2016; Vionnet et al., 2016) are not mentioned in the introduction. Similar studies using other models have also been carried out and are not mentioned in the text. For example, the work done by Bellaire et al. (2011, 2013) and Jamieson and Horton (2015) with the Canadian GEM model to drive the detailed snowpack model SNOWPACK. The authors should mention in the introduction how their work differs from these previous studies and what is their contribution to this field of mountain snow research.

P 2 L 21: what are the reasons behind the selection of the simulation domain in South Norway? Hydropower forecasting? Avalanche hazard forecasting?

Section 2.1: The description of the configuration of Crocus and SURFEX should be more specific. For example, the following points should be clarified: - how many layers are used in the soil models? - how are determined the soil and surface properties (clay and sand fraction, vegetation type, . . . )? - how large is the simulation domain in km and grid points? - how are initialized the soil and snowpack properties (if any snow is present) on 1st September 2014? Did the authors perform a model spin-up?

P 5 L9-10: how many stations are used to generate the gridded precipitation and temperature products in the region? In particular, are these stations covering a similar altitudinal range compared to the stations used for snow depth evaluation?

P 6 L 6: on Fig. 1, it seems that the stations are not covering the area of high elevation of the simulation domain. To illustrate this, point, I recommend the author to add on Fig. 2 the histogram of the distribution of elevation in the simulation domain.

P 6 L 17: are MODIS snow cover data not available for winter 2015/2016? It would be interesting to compare the evolution of simulated and observed snow cover for this winter as well to see if model results are consistent in between the two winters.

P 7 L 10-11: where are located the stations used to illustrate model performance? It would be interesting to see their location on Fig. 1. In particular, it would be interesting to see their locations along the West-East transect. Indeed, we can expect significant differences in terms of precipitation amount and resulting snow accumulation between the western and the eastern side of the domain due to the proximity with the ocean. In this context, elevation is not the only variable that can explain differences of snow depth from one station to another.

P 9 L 7-10: differences of snow depth between GridObs–Crocus and AROME-Crocus are low at Hemsedal II. To support their statement on the best results of GridObs–Crocus compared to AROME-Crocus at this station, I recommend the author to compute bias and RMSE of snow depth at this station for the two simulations.

P 9 L 12-13: what are the reasons behind this under-estimation of temperature? Is it associated to a large difference between the model and the actual terrain height at station location? Can the authors justified that this underestimation is responsible for an overestimation of the proportion of precipitation falling as snow? From my experience, NWP model can present a negative bias of temperature during clear nights in wintertime. However, this bias does not affect the phase of precipitation during precipitation events characterized by overcast conditions.

P 9 L 17: the beginning of winter 2015 at Midtsova is interesting and shows a net underestimation of snow depth by GridObs–Crocus. Is it associated with an underestimation of precipitation in the GridObs or with errors in the phase of precipitation?

P 9 L 29-30: it is surprising to see that the authors have selected a category that cannot be used to classify observations ([-0.5 0.5] cm). If the snow depth does not change from one day to another, what is the corresponding category? I recommend the author to use a central category that can be used to classify both simulation and observation.

P 11 Table 2: it would be interesting to see the RMSE for the different variables as well.

Maybe make two tables if the number of information is too large.

P 11 L6 : Quéno et al. (2016) used the same criteria to define blowing snow days but they used the wind speed measured at the stations instead of the wind speed in the atmospheric forcing. Can the author comment on this choice? How accurate is the forecast wind speed for the different stations used in this study? If the wind speed is measured at some stations measuring snow depth as well, it would be interesting to compare the occurrence of blowing snow days with the two wind data to make sure that forecast wind speed can be used to determine the occurrence of blowing snow days.

P 13 L 20-34: the visual comparison of snow cover patterns proposed on Fig. 9 is useful but it should be complemented by a more quantitative analysis. The author could for example compare the temporal evolution of snow cover area in the observations and in the simulations across different altitudinal bands. Similarity metrics such as the Jaccard index or the confusion matrices could be computed as done in previous studies (e.g. Gascoin et al., 2015; Quéno et al., 2016).

Technical comments

Text

P 1 L 24: remove parenthesis around Bokhorst et al. (2016)

P2 L 20, L31: the correct reference for the Crocus paper is Vionnet et al. (2012). The authors should refer to the final version of the paper and not the discussion version.

P 3 L3 and throughout the rest of the paper: units should be written kg m-2 instead of kg/m2.

P 4 L 16-17: from the 1800 UTC analysis time, are the authors using the 3-8h lead time or the 3-5h lead time? It is not clear since they mention that they use the 0-8 lead time for the 0000 UTC cycle.

P 9 L 4: "Episodes when" instead of "episodes where".

P 9 L 33: " the transport of blowing snow or wind-induced ablation" is not clear and should be rewritten. Maybe : "SURFEX/Crocus in stand-alone mode does not account for wind-induced snow redistribution".

P 12 L2: when snow is present on the ground, maximal surface temperature is 0°C. Please remove "or above".

P 18 L 21: incoming longwave and shortwave radiations are also a key component of the snowpack evolution. Therefore, I recommend the authors to remove the sentence "the two most important variables for snow modeling".

Figure

Figure 1: the name of all the cities on the snapshots from Google Maps are not easy to read and may be removed. Google Maps may not be the most relevant background map.

Figure 3: the axis labels and the legend are too small and hard to read.

Figure 4: it would be very interesting to know the elevation of the model grid point corresponding to the station location. Such information is really relevant to analyse model results (see the general comment on this particular point).

Figure 6 and 7: the size of the markers (squares and diamonds) and of text (legend, axis labels, . . .) is too small on these figures.

References (not included in the initial manuscript):

Carrera, M. L., Bélair, S., Fortin, V., Bilodeau, B., Charpentier, D., & Doré, I. (2010). Evaluation of snowpack simulations over the Canadian Rockies with an experimental hydrometeorological modeling system. Journal of Hydrometeorology, 11(5), 1123-1140.

Bernier, N.B., S. Bélair, B. Bilodeau, and L. Tong, 2011: Near-Surface and Land Surface Forecast System of the Vancouver 2010 Winter Olympic and Paralympic Games.

J. Hydrometeor., 12, 508–530, https://doi.org/10.1175/2011JHM1250.1

Gascoin, S., Hagolle, O., Huc, M., Jarlan, L., Dejoux, J.-F., Szczypta, C., Marti, R., and Sánchez, R.: A snow cover climatology for the Pyrenees from MODIS snow products, Hydrol. Earth Syst. Sci., 19, 2337-2351, https://doi.org/10.5194/hess-19-2337-2015, 2015.

---

## Author Response (AR1)

Reviewer comments are in black, author responses are in *blue Italic*. We will submit a revised manuscript by the end of February 2018. All references to page/line numbers and figures refer to the submitted paper (not the revised manuscript).
Author responses in *red Italic* are added at the time of the submission of the final manuscript.

This article, entitled "Forcing the SURFACE/Crocus snow model with continued hourly meteorological forecasts and gridded observations in southern Norway", has for its main objective to improve numerical prediction of terrestrial snow in Norway. The approach that is presented and examined is based on the used of an external land surface system, SURFEX, that includes the sophisticated snow model CROCUS, and which is driven by a combination of surface analyses (for precipitation and air temperature) and of short-range numerical atmospheric prediction.

This subject is certainly of interest. Better analysis and prediction of terrestrial snow remains one of the great challenges that national environmental prediction centers face. But unfortunately the scope and ambitions of this study are insufficient, in the sense that this kind of work has often been presented in previous papers (which are not cited), and are actually used Operations in several centers. While the results analysis is very well done, with interesting metrics and with interesting discussion, the overall impact (positive) is not Earth shattering.

Unless the authors come up with a major overhaul of the article, maybe based on some of the more specific comments below, this paper should be rejected.

*Author response: Thank you for taking the time to review our paper, it is very much appreciated. The manuscript will be greatly improved thanks to the comments and suggestions from the 2 reviewers.*

*We will address the issues you mention and improve the introduction to discuss previous papers on this topic. We are of the opinion that the originality of our work lies in both the evaluation of the two forcing datasets, and the use of the gridded observations of hourly precipitation and temperature for snow modeling. This dataset has been developed very recently (see Lussana et al, 2017 and 2018, full references below). Most operational snow models for hydrological forecasting in Norway use daily data of precipitation and temperature, while this study was done with hourly data. This is the reason why it is a very interesting dataset to study for hydrological users in Norway. We think that evaluation of the use of gridded observations with a temporal resolution of 1 hour and a spatial resolution of 1 km to force SURFEX/Crocus is both interesting and original.*

*We will answer to the specific comments below (which we have numbered for easier reference), and we will submit a revised manuscript (based on the comments from both reviews) by the end of February 2018.*

*Thanks again for taking the time and effort to review our paper.*
*Kind regards,*
*Hanneke Luijting*

*References to papers mentioned:*

*Lussana, C., Tveito, O., and Uboldi, F.: Three-dimensional spatial interpolation of two-meter temperature over Norway, Quarterly Journal of the Royal Meteorological Society, https://doi.org/10.1002/qj.3208, http://dx.doi.org/10.1002/qj.3208, qJ-17-0046.R2, Accepted Author Manuscript, 2017*

*Lussana, C., Saloranta, T., Skaugen, T., Magnussson, J., Tveito, O. E., and Andersen, J.: seNorge2 daily precipitation, an observational gridded dataset over Norway from 1957 to present days, Earth System Science Data, accepted for publication, 2018*

**Specific comments**: —————

1.  Page 2, line 25: The main objective of the paper is stated as: "The aim was to compare and combine different forcing data sets as input to the SURFEX/Crocus model and validate the computed accumulated snow amounts and snow melt pattern in both Norwegian mountains and lowlands." The authors have to realize that this type of work has been done often. Practically every snow scheme that has been developed in the last few decades have been tested at observational surface stations using atmospheric forcing that are not unlike what is used in this study. Many land data assimilation systems also use snow models forced with a mixture of observations and model predictions to produce terrestrial snow analyses. In order to make this article acceptable for publication, the authors need to describe and include these previous studies in their Introduction, and explain how exactly what they have done contributes in an original manner to advancing this body of research.

    *Author response: Thank you for your suggestion. The introduction will be rewritten and improved in the revised manuscript, following your suggestions. We will include a section in the introduction that better explains the originality of our work compared to previous work, which lies in the evaluation of the two datasets against each other, and the use of gridded observations with a temporal resolution of 1 hour. See also our answer above, and to comments 4 and 8 of review #2.*

    *Update: Update: The introduction has been thoroughly rewritten. We have included many more references to relevant work, more details on the use of NWP data as forcing for snow models, and a section on the originality of our work compared to previous work. The description includes studies carried out for both Norwegian snow conditions as well as for other arctic and alpine areas.*

2.  Page 4, Table 1: Was there any adaptation applied to the atmospheric forcing, e.g., for air temperature, or precipitation phase? My understanding that the atmospheric model that provided some of the forcings was at lower horizontal resolution than the external land surface model. There might be some inconsistencies in terrain height (between model and reality) that could lead to biases.

    *Author response: Thank you for your comment. It is correct that the atmospheric model has a lower resolution (2.5 km) than SURFEX/Crocus (1 km). The forecasts from the atmospheric model were interpolated to a 1 km grid using bilinear interpolation. No downscaling or corrections have been performed to account for the difference in terrain height. We are very much aware of the uncertainty introduced by downscaling the AROME MetCoOp data to 1 km. During the project we discussed the various uncertainties, and found that as a starting point we carry out the simulations as described here.*

*We will add this topic to the discussion (and suggestions for future work) in the revised manuscript. See also our answer to review #2, comment 2.*

*Update:*

*What we should also mention is that the temperature used in AROME-Crocus has a resolution of 500 m, as this is a post-processed, and terrain-adjusted variable from AROME-MetCoOp. We have now clarified this in section 2.3.1.*

*The following text has been added to section 2.3.1*
*"The AROME-MetCoOp temperature used to force SURFEX/Crocus has a grid spacing of 500m. This is a post-processed variable that uses a Kalman filter correction at observation stations, which is then interpolated horizontally using decreasing weights with increasing distance from the station. The temperature is further corrected with a height correction which also takes into account vertical temperature profiles in inversion situations in winter time. The wind speed from AROME-MetCoOp has also been statistically post-processed to represent the maximum wind speed during the last hour. In addition, correction factors are applied to the wind speed depending on wind direction and on the region."*

*The following text has been added to section 4.2:*
*"In this study, the forcing data from AROME-MetCoOp is interpolated from the original 2.5 km resolution to the 1 km resolution required by SURFEX/Crocus, apart from the temperature which is a post-processed, terrain-adjusted variable with a resolution of 500 m. The interpolation of AROME-MetCoOp data over a domain with complex topography could lead to differences in elevation (between AROME-MetCoOp and SURFEX/Crocus), which might lead to bias in for example the precipitation. We believe this might affect the AROME-Crocus experiment more than the GridObs-Crocus experiment, since the two experiments use different data sources. "*

*The following sentence has been added to the Conclusions chapter, in the paragraph about future work:*
*"When using AROME-MetCoOp as forcing data for running SURFEX/Crocus at a resolution higher than 2.5 km, terrain adjustment routines should be applied to the generation of forcing data. In this study we accounted for local terrain effects, by using post-processed AROME-MetCoOp temperature and wind, but this could be extended to other variables."*

3. Page 7, line 8: "... is in good agreement with the observations, ..." I don't know how we could say that from Fig. 3. More generally, more discussion is needed to correctly present and understand what is shown in that figure. For instance, it seems there is a significant bias for snow depth below 100 cm. Also, it might be better to show plot the density of points rather than just the cloud of points.

*Author response: Thank you for your helpful comment. We will add more discussion about what is shown in figure 3 to the revised manuscript. We also agree that figure 3 could be improved, and we will investigate if a density scatter plot(s) would be an improvement.*

*Author response update: Figure 3 has been changed in the revised manuscript, to two seperate density plots (one for AROME-Crocus and one for GridObs-Crocus). Having two plots, and adding the density of points to the plots, makes it easier to understand the data presented. Thank*

*you for this suggestion. The discussion of the plot has also been changed, with more discussion of what is shown in figure 3.*

4. Page 9, line 9: "... matches the observed pattern of increases and decreases more closely than AROME-Crocus." I disagree with that statement... or I would say instead that this improvement is quite marginal. Is it what we should expect in terms of impact for points that do not benefit of having a surface observational station?

   *Author response: To support this statement we have calculated the bias and the RMSE for Hemsedal II: the bias is 25 cm for GridObs-Crocus (RMSE 27 cm) and 30 cm for AROME-Crocus (RMSE: 33 cm). This shows that GridObs-Crocus does perform better than AROME-Crocus at this location, but indeed not by a very large margin. However, Hemsedal II is performing much better than most stations in AROME-Crocus (overall bias for all stations for AROME-Crocus is 42 cm, and RMSE 68 cm). The fact that GridObs-Crocus still outperforms AROME-Crocus also at points that do not benefit of having precipitation measurements (and moreover, a point that performs better than most in AROME-Crocus) is interesting. See also our response to comment 15 from reviewer 2.*

   *Changes in the manuscript:*

   *The sentences "GridObs-Crocus overestimates the snow depth at Hemsedal II, but not to the same extent as AROME-Crocus does. GridObs-Crocus matches the observed pattern of increases and decreases more closely than AROME-Crocus." have been removed.*

   *Instead the following text has been added: "The bias in snow depth at Hemsedal II for the two seasons combined is 25 cm for GridObs-Crocus (RMSE: 27 cm) and 30 cm for AROME-Crocus (RMSE: 33 cm). When compared to the bias (6 cm for GridObs-Crocus and 42 cm for AROME-Crocus) and RMSE (28 cm for GridObs-Crocus and 68 cm for AROME-Crocus) for all stations for the two seasons combined, it shows that Hemsedal II performs better than most stations in AROME-Crocus. For GridObs-Crocus, the bias at Hemsedal II is larger than at most stations, while the RMSE is slightly better. The fact that GridObs-Crocus still outperforms AROME-Crocus even at a station that is not part of the gridded observation dataset is interesting."*

5. Page 16, line 1: "The results are promising." Too vague.

   *Author response: We have removed "The results are promising" from the first sentence, which now reads "Although both experiments are capable of simulating the snow pack over the two winter seasons, there is an overestimation of snow depth in the AROME-Crocus experiment…" (see also our reply to the next question). The same sentence ("The results are promising") has also been removed from the abstract, see question nr 12 under "minor comments".*

6. Page 16, line 1: "Both experiments are capable of simulating the snow pack over the two winter seasons." Based on current and recent scientific and technological achievements in this research area, should we consider this an achievement worth being presented as a conclusion?

   *Author response: Thank you for your comment. This sentence is not meant to be presented as a conclusion (it does not return in the Conclusions chapter either), merely as an introductory*

*sentence for discussing the results in more detail. We first state the obvious, before going into more details about the positive and negative results from both experiments.*

*To make it clearer that it is not meant as a conclusion, we have changed the sentence as follows: "Although both experiments are capable of simulating the snow pack over the two winter seasons, there is an overestimation of snow depth in the AROME-Crocus experiment…"*

7. Page 16, line 16: "... it could still be argued that the GridObs-Crocus are best in locations of the observations that are included in the gridded dataset used to force SURFACE/Crocus." This statement by the authors is not substantiated by evidence in this paper.

   *Author response: Thank you for your comment. This sentence was not meant as a statement that we have evidence for, it was meant to show that we are aware that we might be criticized for evaluating GridObs-Crocus using stations that were part of the gridded precipitation observations used to force GridObs-Crocus. One way to investigate this issue is to have a closer look at the results for Hemsedal II, the only station not included in the gridded precipitation dataset - see our answer to comment 4 (as well as to comment 15 from reviewer 2).*

   *Changes in the manuscript: We have added the following text to the statement: "The only station that was not part of the gridded precipitation dataset is Hemsedal II. The RMSE for GridObs-Crocus at Hemsedal II is 27 cm, about the same as the overall RMSE for all stations for GridObs-Crocus (28 cm). Although this shows that the performance of a station not included in the gridded precipitation dataset is about the same as the performance of stations that are part of this dataset, one station is not enough to draw conclusions about the entire domain."*

8. Page 17, line 28: "... this could lead to an overestimation of the snow cover. This in turn would lead to an overestimation of the snow depth." How does that work? What's the physical link here, to explain this cause and effect?

   *Author response: Thank you for bringing this up. We agree that this was not explained very well. We will improve the discussion of this issue in the revised manuscript.*

   *Update: the text in the discussion has been changed to:*

   *"The SURFEX/Crocus model was originally developed for use in high alpine regions, where there is not a lot of vegetation. In those areas, the assumption of the uniform snow cover is realistic, as there is no interaction with vegetation, but for areas covered with forest and closer to sea level this could lead to an overestimation of the snow cover. When the snow cover is overestimated, the albedo will be too high and this will slow down the snow melt at the end of the season. This might explain the underestimated snow melt in both experiments. "*

**General comments:**

9. The impact of precipitation and air temperature observations on the simulation could be better highlighted with "leave-one-out" experiments.

   *Author response: Thank you for your good suggestion. It would have indeed been interesting to investigate the impact of precipitation and air temperature separately. This was not the focus of*

*our paper however, and it would be too comprehensive to start a new set of experiments and discuss the results in this paper. Instead we will add this topic as a very interesting direction for future work. We will also include more in-depth evaluation of the temperature and precipitation forcing datasets in the revised manuscript.*

*Changes in manuscript: the following sentence has been added to the conclusions: "To investigate the impact of using gridded observations of temperature and precipitation separately, "leave-one-out" experiments could be carried out (two extra experiments where one uses only gridded observations of temperature, and one uses only gridded observations of precipitation, while all other variables come from AROME-MetCoOp)."*

10. **General comment:** The article would gain in quality if a comparison of the results presented in this article would be compared with what is currently available (operationally) in Norway.

*Author response: Thank you for this suggestion. A paper has been submitted by Skaugen et al last month, titled "In search of operational snow model structures for the future - comparing four snow models for 17 catchments in Norway" (full reference below). In this paper, GridObs-Crocus is compared to 3 other models. One of the models is seNorge, which is used operationally by NVE (The Norwegian Water Resources and Energy Directorate). In this study by Skaugen et al, all models use the same gridded dataset of precipitation and temperature as in our study. We have therefore decided to focus on evaluating the use of this gridded dataset as forcing versus using only meteorological forecasts, so that the two papers compliment each other.*

*In the revised manuscript, we will mention this paper by Skaugen et al.*

*Reference: T. Skaugen , H. Luijting, T. Saloranta, D. Vikhamar-Schuler and K. Müller: "In search of operational snow model structures for the future - comparing four snow models for 17 catchments in Norway", Hydrology Research (Submitted December 2017)*

*Update: The reference has been added to the revised manuscript and mentioned in the introduction.*

11. **General comment:** Has there been any tests to evaluate the impact of air temperature and precipitation seperately?

*Author response: Thank you for your comment. We are not entirely sure of the difference (if any) between comment 9 (above) and this comment. In the revised manuscript we will have a stronger focus on the evaluation of the forcing data sets, and we will add more discussion on the temperature and precipitation on the SURFEX/Crocus results in the revised manuscript. We have not looked at the impact of air temperature and precipitation separately, but this would be a very interesting experiment for future work. It has been added to the section about future work, see our answer to comment 9 above.*

**Minor comments**: ——————————

12. Page 1, line 8: "The results are promising." This is too vague.

*Author response: Thank you for your comment. We have removed this sentence from the abstract, as we agree that this is too vague. It was meant as an introduction to the following sentences that describe the results in more detail, but this is not really necessary.*

13. Page 2, line 8: "... include more statistics to capture the physical snow processes". The word "capture" is not appropriate in that sentence.

*Author response: We have replaced the word "capture" with "describe".*

14. Page 2, line 10: "grid spacing" instead of "resolution".

*Author response: We cannot find the word "resolution" on page 2, line 10. The word is used two times on page 2, both in the context of temporal resolution, where we would not want to use grid spacing. On page 4, line 10, we changed horizontal resolution to horizontal grid spacing. We hope this is what was meant here.*

15. Page 2, line 11: "levels" instead of "layers".

*Author response: We again have to assume that the reviewer means page 4, line 11 here. This was changed to "The atmosphere is divided into 65 vertical levels, with the first level at…"*

16. Page 2, line 14: "... the atmospheric part..." this is too vague.

*Author response: This should again be page 4, line 14. The sentence has been rewritten to: "The fluxes computed by SURFEX at the atmosphere–surface interface serve as the lower boundary conditions for the atmosphere within AROME MetCoOp."*

17. Page 7, Fig. 3: The text on this figure is too small, unreadable.

*Author response: The figure has been changed so that the axis labels and legend are much easier to read.*

18. Page 10, Fig. 6: The two colors chosen for this figure are too alike (difficult to distinguish for old eyes like mine...)

*Author response: Thank you for pointing this out. We have changed the color to a lighter blue to make it easier to distinguish the two colors. We wanted to keep blue as this color is used for the GridObs-Crocus results throughout the paper. We have also increased the font size for all text, as well as the size of the markers, in response to a comment (nr 33) from reviewer #2. We hope this makes the plot easier to read.*

19. Page 11, Table 2: The statistics presented in that table are quite interesting, but reading it is a bit tedious. I wonder if there could be another way of arranging the table.

*Author response: Thank you for your comment. The table has been changed: the results for GridObs-Crocus and AROME-Crocus are now placed in columns instead of multiple rows, which makes the table more compact, and hopefully easier to read.*

20. Page 11, line 3: "... exceeds 8 m/s..." this is for the winds at what height, 10 m? This should be mentioned.

*Author response: Yes the wind speed is at 10 m height. The sentence has been changed to clarify this: "Blowing snow days are defined as days during which the wind speed (at a height of 10 m)..."*

21. Page 12, Fig. 7: same comment as Fig. 6 concerning the colors.

*Author response: This plot has been changed in the same way as figure 6, see our reply to comment nr 18 above.*

22. Page 15, Fig. 9: the text is too small, unreadable.

*Author response: This figure has been changed so that all the text (on the axes, axes labels and titles) is larger, and readable.*

**Anonymous Referee #2**

Reviewer comments are in black, author responses are in *blue Italic*. We will submit a revised manuscript by the end of February 2018. All references to page/line numbers and figures refer to the submitted paper (not the revised manuscript).
Author responses in *red Italic* are added at the time of the submission of the final manuscript.

This paper presents an evaluation of the ability of the snowpack model Crocus to simulate snow depth and snow cover in southern Norway for two winters using different atmospheric driving data: (i) short-range high resolution weather forecast generated by the AROME MetCoOP system and (ii) gridded datasets for precipitation and temperature derived from observations. The authors propose an evaluation of model results using snow depth observations collected at 30 stations across the simulation domain and MODIS snow cover data at 500-m grid spacing. Daily snow depth variations are considered as in Quéno et al. (2016) to discuss more in details the physical processes responsible for differences between simulated and observed snow depth.

The subject of this paper is interesting for the snow and mountain hydrology community because of the growing use of high-resolution weather forecast to drive snowpack models in mountainous terrain (e. g. Bellaire et al., 2011, 2013; Bernier et al., 2012; Carrera et al., 2009; Horton and Jamieson, 2016; Quéno et al, 2016; Vionnet et al, 2016). The analysis of results presented here is similar to the studies by Quéno et al (2016) and Vionnet et al. (2016) and reveals consistent and interesting model behavior between the French and the Norwegian mountains. My main comments about this study concern (i) the comparison between the different atmospheric driving datasets, (ii) the interpolation of AROME forecast on the 1-km grid used for Crocus simulations, (iii) the selection of stations for model evaluation and its impact on the analysis of model results and (iv) the originality of this work compared to other studies using AROME and Crocus in the French mountains. These questions need to be clarified prior to publication in TC. They are listed below as general comments followed by more specific and technical comments.

*Author response: Thank you very much for such a detailed review of our paper - we really appreciate your thoughts, comments and suggestions. We will answer to each comment below, which we have numbered for easier reference. We will submit a revised manuscript (based on the comments from both reviews) by the end of February 2018.*

*Thanks again for taking the time and effort to review our paper.*
*Kind regards,*
*Hanneke Luijting*

**General comments**

1. The comparison between simulated snow depth and snow cover using different precipitation and temperature forcing is interesting and illustrate well the strong impact of these variables on simulated snowpack evolution. However, the authors only present the results of snowpack simulations and never compare for example the precipitation forcing in their two experiments; how they differ for different elevation ranges or distance to the sea. Such comparison would be really useful to better understand the differences obtained in the simulated snowpack evolution. For example, Figure 9 shows that the snow cover remains longer at high-elevation in AROME-Crocus compared to GridObs-Crocus. Is it explained by lower precipitation at

high-altitude in the GridObs forcing compared to the AROME forcing? This is not mentioned in Section 3.3 and in the conclusion and should be added to the paper.

*Author response: Thank you very much for this good suggestion. We will focus more on the evaluation of the forcing data set in the revised manuscript. We will include a comparison of the precipitation forcing in the two experiments.*

*Update: An evaluation of the precipitation forcing from the two experiments show that while the amount of rainfall in both experiments is often quite close together, there are big differences in the snowfall amount. AROME-Crocus consistently has larger amounts of snow, often 1.5-2 times as much as GridObs-Crocus, and in some cases more than 3 times as much. The differences are largest on the west side of the domain compared to the east side of the domain. Altitude plays less of a role. It is interesting that the overestimation of total precipitation in AROME-MetCoOp might be limited to the snowfall amount.*

*The following text has been added to section 3.1:*
*"An evaluation of the precipitation forcing data for AROME-Crocus and GridObs-Crocus for the six stations from Fig. 6 reveals that while the rainfall amount is often quite similar in the two experiments, the largest differences are found in the snowfall amount. AROME-Crocus consistently shows a larger amount of snowfall (accumulated over a year) compared to GridObs-Crocus, often about twice as much. The differences are largest for the stations located in the western part of the domain. As described in section 2.2, this region receives the highest amounts of winter precipitation in our study area."*

*The following text has been added to section 3.3:*
*"As discussed previously, the differences between the snowfall amounts from the two precipitation forcing datasets is largest on the west side of the domain; AROME-Crocus receives about twice as much snow compared to GridObs-Crocus in this region. This explains why the differences between GridObs-Crocus and AROME-Crocus in Fig. 10 are also largest in this area.*
*"*

*The following text has been added to section 4.2:*
*"An evaluation of the accumulated precipitation from the two forcing datasets showed that while the rainfall amounts are rather similar, the snowfall amount in AROME-Crocus is much larger compared to GridObs-Crocus, especially at the west side of the domain. This suggests that the overestimation of total precipitation in AROME-MetCoOp might be limited to the snowfall amount."*

2. AROME forecast at 2.5 km are interpolated bilinearly on the 1-km grid used for Crocus simulation. No downscaling is performed to account for the differences between the interpolated terrain height from the 2.5-km grid and the actual terrain height in the 1- km grid. This can potentially lead to large errors in region of complex terrain. For example, the phase partitioning simulated by the AROME cloud microphysical scheme is only valid at the elevation of the grid cell on the 2.5 km grid. A first order correction using a simple lapse rate is required to adjust the phase if the elevation difference is large. Overall, I recommend the author to include simple terrain adjustment routines in their AROME-Crocus simulation to use a meteorological forcing valid on the 1-km grid where are performed the snowpack simulation. This can be done using very simple methods such as the ones used in Bernier et al. (2011).

*Author response: Thank you very much for the considerations and your recommendations about the methodology. We are very much aware of the uncertainty introduced by downscaling the AROME MetCoOp data to 1 km. During the project we discussed the various uncertainties, and found that as a starting point we carried out the simulations as described here.*

*However, we believe your recommendations are very interesting and may be studied and examined in future work and future studies. A 1 km resolution dataset is used because operational snow models used by e.g. the national flood forecast service need this resolution. Still, at this time of the paper revision, it is too comprehensive to rerun the 2 years of Surfex/Crocus simulations with a new forcing dataset. We will however add this topic to the discussion (and suggestions for future work) in the revised manuscript.*

*Update:*

*What we should also mention is that the temperature used in AROME-Crocus has a resolution of 500 m, as this is a post-processed, and terrain-adjusted variable from AROME-MetCoOp. We have now clarified this in section 2.3.1.*

*The following text has been added to section 2.3.1*
*"The AROME-MetCoOp temperature used to force SURFEX/Crocus has a grid spacing of 500m. This is a post-processed variable that uses a Kalman filter correction at observation stations, which is then interpolated horizontally using decreasing weights with increasing distance from the station. The temperature is further corrected with a height correction which also takes into account vertical temperature profiles in inversion situations in winter time. The wind speed from AROME-MetCoOp has also been statistically post-processed to represent the maximum wind speed during the last hour. In addition, correction factors are applied to the wind speed depending on wind direction and on the region."*

*The following text has been added to section 4.2:*
*"In this study, the forcing data from AROME-MetCoOp is interpolated from the original 2.5 km resolution to the 1 km resolution required by SURFEX/Crocus, apart from the temperature which is a post-processed, terrain-adjusted variable with a resolution of 500 m. The interpolation of AROME-MetCoOp data over a domain with complex topography could lead to differences in elevation (between AROME-MetCoOp and SURFEX/Crocus), which might lead to bias in for example the precipitation. We believe this might affect the AROME-Crocus experiment more than the GridObs-Crocus experiment, since the two experiments use different data sources. "*

*The following sentence has been added to the Conclusions chapter, in the paragraph about future work:*
*"When using AROME-MetCoOp as forcing data for running SURFEX/Crocus at a resolution higher than 2.5 km, terrain adjustment routines should be applied to the generation of forcing data. In this study we accounted for local terrain effects, by using post-processed AROME-MetCoOp temperature and wind, but this could be extended to other variables."*

3. Section 4.1 shows that the authors include in their analysis stations with large differences between the model and the actual height at station location (up to 450 m). They did not make a selection of stations based on a maximal value for the difference between the model and the

actual height. In their studies, Vionnet et al (2016) and Quéno et al (2016) used for example a maximal elevation difference of 150 m in absolute value. In this paper, 13 stations among 30 correspond to this criteria. What is the impact of these large elevation differences on the evaluation of model results? For example, for the stations with an elevation difference above 250 m, what is the impact on the evaluation of snow cover duration? What about the wind speed simulated at these stations and used to determine the occurrence of blowing snow days? As mentioned in my previous comment AROME forcing are only interpolated bilinearly on the 1-km grid. Therefore, altitude differences between the station height and the elevation in the interpolated terrain at 1 km from the 2.5-km grid can be potentially even larger. The authors only mention the elevation differences in the discussion (Section 4.1). I think this should be mentioned earlier in the paper; for example in Section 2.3.1 when presenting the snow depth observations. Overall, the effects of these large elevation differences should be better quantified.

*Author response: Thank you for your comment. We used nearly all available snow depth observations from eklima.met.no for our area (except for a few stations that had large gaps in the snow depth data during 2014-2016). If we had been more strict with for example the maximum height difference between the model and the actual height, we would have had too few stations left for the validation analysis, like you already mentioned. We will add a discussion of this issue to section 2.3.1, and we will calculate statistics for the stations with smaller height difference to be able to say something about the impact of the large elevation differences on our results, and add the results to our revised manuscript.*

*Update:*

*The topic of height differences between observations and model grid points has now been added to section 2.3.1, see below.*

*We have calculated statistics for only the stations with a height difference of less than 250 m. This does improve the bias and RMSE, but not by very much. The overall bias for snow depth of GridObs-Crocus remains the same (6 cm), while the RMSE changes from 28 to 25 cm. For AROME-Crocus, the bias changes from 42 to 40 cm, while the RMSE increases from 68 to 71 cm. This does not change anything for our conclusions. For the length of the snow season, the differences are larger, but still the overall conclusions would remain the same, as both experiments change in a similar way. The bias in the length of the snow season for GridObs-Crocus changes from 11 to 8 days for year 1 (RMSE 25 days vs 21 days), and from 8 to 4 days (RMSE 21 vs 18 days) for year 2. For AROME-Crocus the bias changes from 34 to 27 days for year 1 (RMSE 44 vs 37 days), and from 18 to 9 days in year 2 (RMSE: 28 vs 20 days). The changes are largest for AROME-Crocus (7-9 days improvement, versus 3-4 days for GridObs-Crocus), but GridObs-Crocus still outperforms AROME-Crocus at all times for this variable. We have added a short version of this quantification to the discussion chapter, see below.*

*Changes in manuscript:*

*The following text has been added to 2.3.1:*
*"In a domain with deep valleys and high mountains, it is difficult to match the elevation of the weather stations with the nearest grid point in the SURFEX/Crocus experiments. As there were only 30 stations with high quality snow depth observations in the domain, it was decided not to*

*filter out stations based on these height differences. This issue is discussed in more detail in section 4.1 in the Discussion chapter. "*

*The following text has been added to 4.1:*
*"It could be argued that SURFEX grid points with a large height difference compared to the corresponding station should not be used for verification purposes. However, when calculating bias and RMSE values only for the stations with a height difference below 250 m, this did not have a large impact on the results. With a few exceptions, the bias and RMSE are lower when excluding stations with a large height difference, but the differences in overall bias are only ± 3 cm, which does not change any of the conclusions in this study."*

4. The simulation framework and evaluation methods presented in this paper are very similar to the ones used by Vionnet et al. (2016) and Quéno et al. (2016) who used AROME to drive Crocus snowpack simulations in the French Alps and the Pyrenees. It is interesting to see that similar results are obtained in a different mountainous environment. However, the author need to better insist on the originality of their study compared to these previous work.

*Author response: We are of the opinion that the originality of our work lies in both the evaluation of the two forcing datasets, and the use of the gridded observations of hourly precipitation and temperature for snow modeling. This gridded dataset has been developed very recently (see Lussana et al, 2017 and 2018, full references below). Most operational snow models for hydrological forecasting in Norway use daily data of precipitation and temperature, while this study was done with hourly data. This is the reason why it is a very interesting dataset to study for hydrological users in Norway. We think that evaluation of using gridded observations with a a temporal resolution of 1 hour and a spatial resolution of 1 km to force SURFEX/Crocus is both interesting and original.*

*We will include a section in the introduction of the revised manuscript that better explains the originality of our work compared to previous work, including work done by Vionnet et al. (2016) and Quéno et al. (2016).*

*References to papers mentioned:*
*Lussana, C., Tveito, O., and Uboldi, F.: Three-dimensional spatial interpolation of two-meter temperature over Norway, Quarterly Journal of the Royal Meteorological Society, https://doi.org/10.1002/qj.3208, http://dx.doi.org/10.1002/qj.3208, qJ-17-0046.R2, Accepted Author Manuscript, 2017*

*Lussana, C., Saloranta, T., Skaugen, T., Magnusson, J., Tveito, O. E., and Andersen, J.: seNorge2 daily precipitation, an observational gridded dataset over Norway from 1957 to present days, Earth System Science Data, accepted for publication, 2018*

*Update: The introduction has been thoroughly rewritten. We have included many more references to relevant work, more details on the use of NWP data as forcing for snow models, and a section on the originality of our work compared to previous work. The description includes studies carried out for both Norwegian snow conditions as well as for other arctic and alpine areas.*

5. It would have been interesting to see additional experiments. For example the authors used a succession of forecast from +3 to +8 to drive Crocus. Vionnet et al. (2016) and Queno et al

(2016) combined daily forecast from +6 to +29 issued at 00 UTC to drive Crocus. The impact of these choices on model results has never been discussed and it would be an interesting contribution.

*Author response: We chose to use +3 to +8 to avoid the first hours of spinup of the model, while making use of all available model runs. We believe this is an advantage compared to using +6 to +29 from only the 00 UTC run. Model errors increase with lead time, and our aim was to use the best available model data. A study on the impact of using different lead times from an atmospheric model to force SURFEX/Crocus would certainly be interesting, but this is beyond the scope of this paper. Our aim was to compare the two forcing datasets. Lead time comparisons would only be relevant for the AROME-Crocus experiment and not for GridObs-Crocus. The gridded observation dataset has a temporal resolution of one hour, and for a fair comparison, we used the best available forcing data from the AROME-MetCoOp model, and this meant using all 4 daily model runs with the shortest possible lead times.*

*Changes in manuscript: the following sentence has been added to 2.2.1: "These lead times were chosen to avoid the first hours of a cycle when the model might have spin-up issues, and to make use of all available cycles with the shortest possible lead time (since model error increases with lead time, see for example Homleid and Tveter (2016))."*

6. Similarly, the authors mentioned at the end of their paper (P 19 L7-8) the potential importance of blowing snow sublimation for the high-altitude part of their domain. I recommend the authors to carry out an experiment where they test the impact of the parametrization of sublimation loss during blowing snow events implemented in Crocus. The authors could discuss the impact in terms of snow cover duration and compare it with MODIS images. Overall, these additional experiments would bring interesting insights and strengthen the discussion section which is so far very similar to the discussions in Vionnet et al. (2016) and Quéno et al. (2016).

*Author response: Thank you for this suggestion. We will perform a 1D experiment that tests the impact of the parameterization of sublimation loss during snow events, and discuss the results of this experiment in the revised manuscript.*

*Update: We have carried out 1D experiments using the GridObs-Crocus setup for two locations: Midtstova and Hemsedal II. As expected, the snow depth is decreased when the option for sublimation loss is turned on, and the snow cover duration is decreased. Where the snow depth is overestimated, this brings a slight improvement, but when the snow depth and season length is already underestimated, this does not improve the results.*

*The following text has been added to the Results chapter, along with a new figure (8)::*

*"SURFEX/Crocus does have an option to run with sublimation in case of snowdrift. This option has been tested for two stations from Fig. 4: Midtstova and Hemsedal II. In this experiment, SURFEX/Crocus was run twice in 1D mode for these 2 locations: one experiment with identical settings as GridObs-Crocus, and one nearly identical with the exception of the option for sublimation in case of snowdrift (GridObs-Crocus+BS). The results are shown in Fig. 8. For both locations, the snow depth in GridObs-Crocus+BS is decreased, as expected. For Hemsedal II, this reduction is an improvement compared to the GridObs-experiment, which overestimated the snow depth. The bias in GridObs-Crocus was +16 cm for Hemsedal II, which has improved to*

*+10 cm in GridObs-Crocus+BS. For Midtstova, GridObs-Crocus underestimates the snow depth for most of the two winter seasons (bias: -5 cm), and this underestimation is significantly larger in the GridObs+BS experiment (-28 cm). "*

*The following text has been added to the discussion:*

*"There is an option in the SURFEX/Crocus model to calculate the rate of sublimation in case of snowdrift, which results in a loss of snow. This option was tested for two stations in his study, using the GridObs-Crocus forcing dataset. As expected, this resulted in a decrease in snow depth and a decrease in season length. As GridObs-Crocus already underestimates the snow depth and snow cover, this is not an improvement."*

*The following sentence has been added to the conclusions:*

*"Using the option in SURFEX/Crocus of running with sublimation in case of snowdrift is not enough to address this issue."*

**Specific comments**

7. **Abstract**: The abstract is rather vague and should present some precise figures such as the overall snow depth bias for the two experiments and the number of stations used for model evaluation. A L11-13, the authors mention the assimilation of snow depth data directly into Crocus. This topic is mentioned here but never discussed in the paper. If the authors want to keep this sentence in the abstract, they need at least to discuss more the assimilation of punctual snow depth data in distributed snowpack simulations in the discussion part.

   *Author response: Thank you for your comment. We have included the overall bias and RMSE of the two experiments to the abstract, and have added the number of stations used for the evaluation. Concerning the assimilation of snow depth, we actually do briefly discuss this topic in the first paragraph of the discussions chapter (page 16, line 5). Our argument is that when errors accumulate during the snow season (due to the overestimation of snow in AROME-Crocus), one solution would be to assimilate observed snow depth into Crocus. We agree however that this topic does not belong in the abstract, and have changed the abstract to reflect this.*

8. **Introduction**: The current introduction of the paper does not described well enough the context of the study and the scientific questions the authors are investigating. For example, the authors never mention the growing use of high resolution NWP forecast to drive detailed snowpack model in mountainous terrain and the limitations associated with these systems. In particular, previous studies using AROME forecasts to drive Crocus in the French Alps and the Pyrenees (Quéno et al., 2016; Vionnet et al., 2016) are not mentioned in the introduction. Similar studies using other models have also been carried out and are not mentioned in the text. For example, the work done by Bellaire et al. (2011, 2013) and Jamieson and Horton (2015) with the Canadian GEM model to drive the detailed snowpack model SNOWPACK. The authors should mention in the introduction how their work differs from these previous studies and what is their contribution to this field of mountain snow research.

   *Author response: Thank you for your comments. We will improve the introduction in the revised manuscript, taking into account your suggestions and discussing previous studies. See also our*

*answer to comment 4 in this review regarding the originality of our work and our contribution to mountain snow research.*

*Update: the introduction has been rewritten, and includes much more discussion on previous work, the originality of our work compared to these studies, and how we contribute to the field of mountain snow research. All the mentioned studies, and many more, have been discussed in the updated introduction. The growing use of high resolution NWP forecasts and their limitations has also been discussed. We believe the context of the study and our scientific questions are now more clearly described.*

9. P 2 L 21: what are the reasons behind the selection of the simulation domain in South Norway? Hydropower forecasting? Avalanche hazard forecasting?

*Author response: We have replaced the sentence "We selected a west-east transect in a mountainous area of South Norway as the study area." with a more detailed explanation of the choice of domain: "The evaluation was done as a part of several research projects within hydropower and flood forecasting. The domain was chosen to cover the mountains in southern Norway and to include a cross-section from west to east that crosses the watershed in this region. The domain also includes several catchment areas that are of interest to hydropower companies."*

10. Section 2.1: The description of the configuration of Crocus and SURFEX should be more specific. For example, the following points should be clarified: - how many layers are used in the soil models? - how are determined the soil and surface properties (clay and sand fraction, vegetation type, . . . )? - how large is the simulation domain in km and grid points? - how are initialized the soil and snowpack properties (if any snow is present) on 1st September 2014? Did the authors perform a model spin-up?

*Author response: Thank you for this helpful comment. We will improve the description of the configuration of our SURFEX/Crocus simulations by including the details you mention, and clarify that no snow is present on 1st of September 2014 (as well as on 1st September 2015).*

*Update - the following text has been added to section 2.1:*

*"In this study, the soil has 14 layers. A force-restore method with 3 layers for hydrology was used for soil discretization and physics within ISBA-DIF. The HSWD (Harmonized World Soil Database) 1 km resolution database for soil texture (FAO/IIASA/ISRIC/ISS-CAS/JRC, 2012) was used for the soil properties."*

*"These dates were chosen because the hydrological year starts on 1 September, and at that time there is normally no snow in the mountains. In this study, we start a new simulation on 1 September, with no snow present, and with default values for soil properties, for both 2014/2015 and 2015/2016. ."*

*The following text has been added to section 2.2 (a new section about the study area):*

*"The domain covers nearly 20.000 km2 (111 x 175 km), and contains 100 x 330 grid points."*

11. P 5 L9-10: how many stations are used to generate the gridded precipitation and temperature products in the region? In particular, are these stations covering a similar altitudinal range compared to the stations used for snow depth evaluation?

*Author response: The number of stations that are part of the dataset is variable with time (new stations are added, sometimes stations are closed down). The number of stations for out SURFEX/Crocus domain: 20-30 stations for hourly precipitation, 90-100 stations for daily precipitation and 70-100 stations for temperature. These stations do cover a similar altitudinal range compared to the stations used in the snow depth elevation: for precipitation the highest station is at 1210 masl., for temperature there are higher stations available with the highest at 1390 masl.*

*The following text has been added to 2.2.2: "The number of stations that are included in the gridded dataset is not constant (new stations are added, sometimes stations are closed down). The numbers of stations within the SURFEX/Crocus domain are: 20-30 stations for hourly precipitation, 90-100 stations for daily precipitation and 70-100 stations for temperature. Stations just outside the domain are included in this estimate as they are used in the interpolation and are therefore part of the gridded dataset used in this study. "*

12. P 6 L 6: on Fig. 1, it seems that the stations are not covering the area of high elevation of the simulation domain. To illustrate this, point, I recommend the author to add on Fig. 2 the histogram of the distribution of elevation in the simulation domain.

*Author response: This is a well known situation in Norway that most weather stations are located at low elevations (at the bottom of valleys), and there are too few stations high up in the mountains. We have included all the available high quality stations that are located within the domain and observe snow depth, from eklima.met.no (a few stations were discarded due to snow depth observations missing for a long period of time within the two years of this study). We will add a histogram of the model elevations to figure 2 in the revised manuscript, and discuss this topic in more detail in section 2.3.1.*

*The new figure 2 shows a histogram of the elevation of grid points within the SURFEX domain (as well as the elevation of the 30 weather stations). The following text has been added to 2.3.1:*

*"The elevation of the grid points in the SURFEX domain is also shown in Fig. 2, which shows a typical issue with the location of weather stations, particularly in the Norwegian mountains: there are many stations located at low elevations (at the bottom of valleys), and very few stations at high elevations."*

13. P 6 L 17: are MODIS snow cover data not available for winter 2015/2016? It would be interesting to compare the evolution of simulated and observed snow cover for this winter as well to see if model results are consistent in between the two winters.

*Author response: Thank you for your suggestion. Unfortunately, the processed MODIS snow cover images for 2015/2016 are not available at this point. It would indeed be interesting to do the same comparison for a second winter season, if images had been available.*

14. P 7 L 10-11: where are located the stations used to illustrate model performance? It would be interesting to see their location on Fig. 1. In particular, it would be interesting to see their locations along the West-East transect.

*Author response: Thank you for this very good suggestion. We have added the locations of the stations used in Fig. 4 with a blue color and an indication for the name of the station in Fig. 1. This shows that the 6 stations are quite evenly spread over the domain from west to east. The caption of Fig. 4 has been changed to point to Fig 1. for the locations of the 6 stations, and the same has been done with the text in 3.1 which refers to the 6 stations.*

Indeed, we can expect significant differences in terms of precipitation amount and resulting snow accumulation between the western and the eastern side of the domain due to the proximity with the ocean. In this context, elevation is not the only variable that can explain differences of snow depth from one station to another.

*Author response: This is correct, and the climatology of Norway means a lot more precipitation falls on the western part of the watershed than on the eastern side. We will add climatology information to the study area description to clarify this.*

*Update: the following text has been added to the new section 2.2 Study Area:*

*"Due to the watershed and the prevailing weather patterns, there is a large gradient in precipitation amount over the domain. The far western parts of the domain receive on average around 1500 mm of precipitation during a winter season, while the eastern parts only receives 100-300 mm (Hanssen- Bauer et al, 2015). The western part of the domain has a maritime climate while the eastern part has a more inland climate, which means the average temperature during winter is higher at the western part of the domain (around or just below 0 \degree Celsius), compared to the eastern side (around -10 \degree Celsius) (Hanssen- Bauer et al, 2015). This means the gradient in average snowfall amount is not as large as the gradient in precipitation amount, but the western part of the domain still receives significantly more snow than the eastern part (Hanssen- Bauer et al, 2015). "*

15. P 9 L 7-10: differences of snow depth between GridObs–Crocus and AROME-Crocus are low at Hemsedal II. To support their statement on the best results of GridObs– Crocus compared to AROME-Crocus at this station, I recommend the author to compute bias and RMSE of snow depth at this station for the two simulations.

*Author response: Thank you for this good suggestion. The bias at Hemsedal II for GridObs-Crocus is 25 cm vs 30 cm for AROME-Crocus. The RMSE at Hemsedal II for GridObs-Crocus is 27 cm and the RMSE for AROME-Crocus is 33 cm. These differences are not very large, but GridObs-Crocus does perform better than AROME-Crocus at this station. It is also worth keeping in mind that the overall RMSE for GridObs-Crocus (for all stations and for the 2 years combined) is 28 cm (bias: 6 cm) and for AROME-Crocus 68 cm (bias: 42 cm). This means that Hemsedal II is performing much better than most stations in AROME-Crocus. The interesting part is that precipitation is not measured at Hemsedal II, and therefore this station is more representative for the performance of GridObs-Crocus outside the stations that are part of the gridded precipitation forcing dataset. It is therefore interesting that GridObs-Crocus still performs better than AROME-Crocus at this location.*

*Changes in the manuscript: The sentences "GridObs-Crocus overestimates the snow depth at Hemsedal II, but not to the same extent as AROME-Crocus does. GridObs-Crocus matches the observed pattern of increases and decreases more closely than AROME-Crocus." have been removed.*

*Instead the following text has been added:*

*"The bias in snow depth at Hemsedal II for the two seasons combined is 25 cm for GridObs-Crocus (RMSE: 27 cm) and 30 cm for AROME-Crocus (RMSE: 33 cm). When compared to the bias (6 cm for GridObs-Crocus and 42 cm for AROME-Crocus) and RMSE (28 cm for GridObs-Crocus and 68 cm for AROME-Crocus) for all stations for the two seasons combined, it shows that Hemsedal II performs better than most stations in AROME-Crocus. For GridObs-Crocus, the bias at Hemsedal II is larger than at most stations, while the RMSE is slightly better. The fact that GridObs-Crocus still outperforms AROME-Crocus even at a station that is not part of the gridded observation dataset is interesting."*

16. P 9 L 12-13: what are the reasons behind this under-estimation of temperature? Is it associated to a large difference between the model and the actual terrain height at station location? Can the authors justified that this underestimation is responsible for an overestimation of the proportion of precipitation falling as snow? From my experience, NWP model can present a negative bias of temperature during clear nights in wintertime. However, this bias does not affect the phase of precipitation during precipitation events characterized by overcast conditions.

    *Author response: Thank you for your comment and thoughts on this issue. Midtstova is located at 1297 m in SURFEX/Crocus (this information is now included in figure 4, see comment 32), which is a difference of 135 m with the actual height (1162 m). This is not a very large difference compared to the other stations, so we do not think the underestimation of temperature is related mainly to this difference in height. We will investigate the role of the underestimated temperature on the overestimated snow depth, and discuss this further in the revised version of the manuscript. We will make a case study about Midtstova to further investigate this issue, see also our reply to comment 17 below.*

    *Update: During the snow accumulation season the temperature at Midtstova is mostly well below freezing level. There are a few episodes each winter with temperatures just above zero, where the underestimated temperature in AROME-MetCoOp means the precipitation during those episodes comes as snow instead of rain, but these do not add up to large amounts.*

    *The following text has been added to section 3.1:*

    *"In the forcing data for Midtstova we find a bias of -1.5 degrees for AROME-Crocus, compared to -0.8 degrees for GridObs-Crocus. This bias is larger than the overall bias for all nine stations measuring temperature: -0.5 degree for AROME-Crocus and -0.2 degree for GridObs-Crocus. During the snow accumulation season the temperature at Midtstova is mostly well below freezing level. There are a few episodes each winter with temperatures just above zero, where the underestimated temperature in AROME-MetCoOp means the precipitation during those episodes comes as snow instead of rain, but these do not add up to large amounts. "*

17. P 9 L 17: the beginning of winter 2015 at Midtsova is interesting and shows a net underestimation of snow depth by GridObs–Crocus. Is it associated with an underestimation of precipitation in the GridObs or with errors in the phase of precipitation?

*Author response: Thank you for your suggestion. This is indeed an interesting episode to investigate further. In the revised manuscript, we will "zoom in" to this episode at Midtstova and investigate what's going on in the forcing data (precipitation, temperature/precipitation phase).*

*Update: The case study revealed that the underestimation of snow depth at Midtstova in the beginning of the winter 2014-2015 is due to underestimation of the precipitation in GridObs-Crocus. There were no issues with the phase of precipitation, as the temperature is below zero for almost the entire period. Further investigation revealed that there is no hourly precipitation data for Midtstova between 27 October 2014 and 26 January 2015 due to a broken sensor. That means that the precipitation at Midtstova will be an interpolated value using surrounding stations during that period in the GridObs forcing, and this might explain why the precipitation is underestimated.*

*The following text has been added to section 3.1:*
*"GridObs-Crocus shows much more realistic results for Midtstova, although there is an underestimation of snow depth during the first part of the 2014-2015 winter. From 27 October 2014 until 26 January 2015, the precipitation sensor at Midtstova was out of order, and the forcing from GridObs-Crocus for Midtstova will therefore be represented by interpolated values from surrounding stations, which might explain the underestimation."*

18. P 9 L 29-30: it is surprising to see that the authors have selected a category that cannot be used to classify observations ([-0.5 0.5] cm). If the snow depth does not change from one day to another, what is the corresponding category? I recommend the author to use a central category that can be used to classify both simulation and observation.

*Author response: This was originally chosen because when we use a category of ([-1 1] cm, a lot of cases fall within this category which then dominates the plot. We agree however that it is not practical to use a category that excludes observations. We will change the central category and include a new figure 5 to the revised manuscript.*

*Update: The [-0.5 0.5] cm category has been replaced by [-1 1] cm in the new Fig. 5. With the logarithmic y axis, this is not a problem at all. Thank you for pointing this out!*

19. P 11 Table 2: it would be interesting to see the RMSE for the different variables as well. Maybe make two tables if the number of information is too large.

*Author response: We have added a new table (table 3) which summarizes the RMSE for the different variables. Note that the layout of table 2 has changed in response to a comment from reviewer #1, and therefore table 2 was changed in the same way.*

*Changes in manuscript: table 3 was added, and the following text was added to section 3.1: "Table 3 summarizes the RMSE over all stations for the two winter seasons. The RMSE values are significantly larger for AROME-Crocus (compared to GridObs-Crocus) for nearly all variables, except for the date of maximum snow depth for 2015-2016. "*

20. P 11 L6 : Quéno et al. (2016) used the same criteria to define blowing snow days but they used the wind speed measured at the stations instead of the wind speed in the atmospheric forcing. Can the author comment on this choice? How accurate is the forecast wind speed for the different stations used in this study? If the wind speed is measured at some stations measuring snow depth as well, it would be interesting to compare the occurrence of blowing snow days with the two wind data to make sure that forecast wind speed can be used to determine the occurrence of blowing snow days.

*Author response: We chose to use model data because only 6 out of the 30 stations measure wind speed. The bias of the forecasted maximum wind speed is 0.3 m/s, which means a slight overestimation of the maximum wind speed by the model. When comparing blowing snow days derived from the observed wind speed at those 6 stations with the forecasted wind speeds from the same 6 stations we find that the model is correct in 94% of the cases (for blowing snow days and non-blowing snow days), with a hit rate of 0.86 (correctly identifying blowing snow days) and a false alarm rate of 0.04 (model data indicates a blowing snow day while observations don't). From this, we conclude that we can use the forecasted wind speed to determine the occurrence of blowing snow days.*

*Changed in the manuscript: we've added the following text to section 3.1 : "The modeled wind speed is used because only 6 out of 30 stations used in this study observe wind speed. When comparing the forecasted maximum wind speed from AROME-MetCoOp with the observed maximum wind speed from these 6 stations, we find a slight overestimation by AROME-MetCoOp (a bias of 0.3 m/s). Blowing snow days and non-blowing snow days are correctly identified in 94% of all days, with a hit rate of 0.86 and a false alarm rate of 0.04."*

21. P 13 L 20-34: the visual comparison of snow cover patterns proposed on Fig. 9 is useful but it should be complemented by a more quantitative analysis. The author could for example compare the temporal evolution of snow cover area in the observations and in the simulations across different altitudinal bands. Similarity metrics such as the Jaccard index or the confusion matrices could be computed as done in previous studies (e.g. Gascoin et al., 2015; Quéno et al., 2016).

*Author response: Thank you for your helpful suggestion. We will perform a quantitative analysis on the snow cover patterns, which will be included and discussed in the revised manuscript.*

*Update: We have calculated the Jaccard index for the SCA images from Fig. 10 (Fig. 9 in the original manuscript). This confirms our visual conclusions that AROME-Crocus performs better in terms of snow-covered area.*

*A new table has been added:*

| Jaccard Index | GridObs-Crocus | AROME-Crocus |
|---|---|---|
| 15 March 2015 | 0.92 | 0.99 |
| 20 April 2015 | 0.82 | 0.94 |
| 15 May 2015 | 0.65 | 0.82 |
| 04 July 2015 | 0.19 | 0.68 |

**Table 4.** Jaccard index for the snow covered areas shown in Fig. 10. A score of 1 means the image perfectly matches the MODIS image, a score of 0 means there is no overlap between the image from the experiment compared to the MODIS image.

*The following text has been added to section 3.3:*
*"Table 4 shows the Jaccard indices for the images from Fig. 10. The Jaccard index was also used by for example Queno et al (2016). It is a similarity index applied to the snow cover images which were remapped onto the same grid (which means that the snow cover from the MODIS images used to calculate the Jaccard index has a lower resolution than the one shown in Fig. 10). The Jaccard index is calculated as $J(X,Y) = |X\ \cup\ Y| / |X\ \cap\ Y|$, where X and Y are the simulated and observed snow cover, respectively. The number of grid points that are snow-covered in both SURFEX/Crocus and in the MODIS image is divided by the total amount of snow-covered grid points (in either SURFEX/Crocus or MODIS). When the Jaccard index equals 1, there is a perfect match between snow-covered grid points, and when the Jaccard index equals 0, there is no match at all. Table 4 shows that AROME-Crocus consistently has higher Jaccard indices compared to GridObs-Crocus. The indices decrease (for both experiments) during the melt season. "*

**Technical comments**

**Text**

22. P 1 L 24: remove parenthesis around Bokhorst et al. (2016)

    *Author response: Parenthesis are removed from this reference.*

23. P2 L 20, L31: the correct reference for the Crocus paper is Vionnet et al. (2012). The authors should refer to the final version of the paper and not the discussion version.

    *Author response: The reference has been corrected to the final version from 2012.*

24. P 3 L3 and throughout the rest of the paper: units should be written kg m-2 instead of kg/m2.

    *Author response: This has been changed throughout the paper, also for other units and for example for table 1.*

25. P 4 L 16-17: from the 1800 UTC analysis time, are the authors using the 3-8h lead time or the 3-5h lead time? It is not clear since they mention that they use the 0-8 lead time for the 0000 UTC cycle.

*Author response: Thank you for pointing this out. From the 18 UTC analysis time we indeed use the 3-5 hour lead time.*

*The sentence has been changed to "Forcing for our study is taken from the 4 main cycles, with successive 3-8h lead time (0-8h lead time for the 0000 UTC cycle, and 3-5h lead time for the 1800 UTC cycle) forecasts combined into a forcing file for each day."*

26. P 9 L 4: "Episodes when" instead of "episodes where".

    *Author response: Changed to "Episodes when"*

27. P 9 L 33: " the transport of blowing snow or wind-induced ablation" is not clear and should be rewritten. Maybe : "SURFEX/Crocus in stand-alone mode does not account for wind-induced snow redistribution".

    *Author response: Thank you for your suggestion. The sentence has been changed to: "SURFEX/Crocus in stand-alone mode does not account for wind-induced snow redistribution, which can be a large contributor to strong decreases in snow depth."*

28. P 12 L2: when snow is present on the ground, maximal surface temperature is 0°C. Please remove "or above".

    *Author response: Changed to "Melting snow days are defined as days when the surface temperature of the snow is 0 °Celsius."*

29. P 18 L 21: incoming longwave and shortwave radiations are also a key component of the snowpack evolution. Therefore, I recommend the authors to remove the sentence "the two most important variables for snow modeling".

    *Author response: Changed to "important variables for snow modeling"*

**Figure**

30. Figure 1: the name of all the cities on the snapshots from Google Maps are not easy to read and may be removed. Google Maps may not be the most relevant background map.

    *Author response: Thank you for your comment. We have added the names of Oslo and Bergen in bigger font on the map. The elevation map is more relevant than the Google overview map, and we have changed figure 1 to better reflect this: the elevation map is now much larger than the Google map. As a quick overview of where the domain is located in Norway we believe the small Google overview map is sufficient.*

31. Figure 3: the axis labels and the legend are too small and hard to read.

    *Author response: The figure has been changed so that the axis labels and legend are much easier to read.*

32. Figure 4: it would be very interesting to know the elevation of the model grid point corresponding to the station location. Such information is really relevant to analyse model results (see the general comment on this particular point).

*Author response: The elevation of the model grid point has been added to the figure, in parentheses after the station elevation. The caption of the figure has been modified to reflect this: "The altitude of the station is indicated above each plot, with in parentheses the elevation of the grid point in SURFEX/Crocus."*

33. Figure 6 and 7: the size of the markers (squares and diamonds) and of text (legend, axis labels, . . .) is too small on these figures.

*Author response: We have changed the size of the markers, and increased the font size of all text for both figures. The color of the dark blue has also been changed to a lighter blue, in response to a comment by reviewer #1 that the two colors (black and dark blue) were hard to distinguish.*

**References** (not included in the initial manuscript):

34. Carrera, M. L., Bélair, S., Fortin, V., Bilodeau, B., Charpentier, D., & Doré, I. (2010). Evaluation of snowpack simulations over the Canadian Rockies with an experimental hydrometeorological modeling system. Journal of Hydrometeorology, 11(5), 1123- 1140.

Bernier, N.B., S. Bélair, B. Bilodeau, and L. Tong, 2011: Near-Surface and Land Surface Forecast System of the Vancouver 2010 Winter Olympic and Paralympic Games. J. Hydrometeor., 12, 508–530, https://doi.org/10.1175/2011JHM1250.1

Gascoin, S., Hagolle, O., Huc, M., Jarlan, L., Dejoux, J.-F., Szczypta, C., Marti, R., and Sánchez, R.: A snow cover climatology for the Pyrenees from MODIS snow products, Hydrol. Earth Syst. Sci., 19, 2337-2351, https://doi.org/10.5194/hess-19-2337-2015, 2015.

*Author response: Thank you for your suggestions. We will add these references to the revised manuscript.*

*Update: the first two references have been added to the revised manuscript and discussed in the introduction. The third reference focuses on the snow maps derived from remote sensing data, while we focus on snow modeling, which is why we have not included this reference.*

[revised manuscript text omitted]

---

## Author Response (AR2)

*Author response: Thank you very much for another detailed review of our paper - we really appreciate your thoughts, comments and suggestions. We will answer to each comment below, which we have numbered for easier reference. We will submit a revised manuscript (based on the comments from the reviewer and from the editor) by early June 2018.*

*Thanks again for taking the time and effort to review our paper.*
*Kind regards, on behalf of all authors,*
*Hanneke Luijting*

*Note: Reviewer comments are in black, author responses are in blue Italic. All references to page/line numbers and figures refer to the latest submitted (revised) paper (not to the revised manuscript).*

Review of the revised version of the paper "Forcing the SURFEX/Crocus snow model with combined hourly meteorological forecast and gridded observations in southern Norway" by H. Luijting et al. submitted to The Cryosphere

General comments

1. During the revision process, the authors have made several changes to the initial manuscript including:
   • An extended introduction that better presents the context of the paper with respect to previous studies and that also gives an overview of the current status of snowpack modeling in Norway for various applications.
   • A better description of the study area and the meteorological forcing provided to Crocus.
   • 1-D simulations to test the impact of blowing snow sublimation on snowpack evolution in Crocus
   • A quantification of the agreement between modeled snow cover and snow cover observed retrieved from MODIS.
   These changes have improved the quality of the manuscript.

   *Author response: Thank you very much, we are glad to hear that you find the quality of the manuscript improved, thanks to all the helpful questions and suggestions by the reviewers!*

2. On the other hand, the authors have chosen not to include any additional 2D simulations as suggested by the two reviewers. This would have potentially increased the quality of the paper and provided interesting results for the community. The computational cost cannot be the only argument considered when deciding to rerun a simulation or not. In particular, my main concern regarding the simulations presented in the revised version of the paper is related to the simulation driven by AROME-MetCoop. In this simulation, air temperature is taken from a post-processed gridded temperature at 500 m whereas precipitation (rainfall and snowfall) are

obtained from AROME at 2.5 km grid spacing. Is the precipitation phase modified based on the high-resolution post-processed temperature data? This is never mentioned in the paper. It can have a impact in the western side of the domain where temperatures can be closed to zero during precipitation events occurring in wintertime. An error in precipitation phase can have a strong impact on the snowpack evolution. If this correction of precipitation phase has not been carried out, I strongly recommend the authors to rerun the Crocus simulation driven by AROME-MetCoop with consistent precipitation fields.

*Author response: Thank you for these comments. We have now carried out a new 2D experiment according to your recommendations. We have rerun AROME-Crocus for the two winter seasons, for the whole domain, by modifying the way the precipitation phase was computed. In our first manuscript, we used the snowfall and rainfall computed by the microphysics of the atmospheric model AROME-MetCoOp. These 2.5 km resolution predictions were used equally for all the 1 km grid cells located inside the 2.5 km (without any other terrain adjustment). In the new experiment, we have computed precipitation phase according to your suggestions. We use the precipitation amounts predicted by AROME-MetCoOp at 2.5 km, but compute the precipitation phase using the temperature threshold of + 0.5 degrees Celsius and the actual temperature from the 500 m high-resolution post-processed temperature data. For each 1 km grid cell, we then compare the temperature from the post-processed temperature grid with the temperature threshold to determine rainfall or snowfall. The same temperature threshold was also used for the GridObs-Crocus experiment.*

*We have updated the entire paper with the new results. We have also improved the description of the forcing data for the AROME-Crocus runs, both in the text (at 2.3.1) and in table 1. We have also added a section in the discussion of the forcing data, concerning different ways of computing precipitation phase from NWP data. The two experiments covers both of our simulations (AROME-MetCoOp's own simulation of rainfall and snowfall, and the new experiment in this revision).*

3. The quality of the discussion should also be improved prior to publication. In the conclusion and at the beginning of the discussion section, the authors explain that AROME-MetCoop-Crocus provides the best representation of the spatial distribution of snow cover during the melting season whereas it gives worse results than GridObs Crocus when evaluated at stations measuring snow depth. I recommend the authors to better discuss the reason for these differences when estimating the performances of AROME-MetCoop-Crocus. Is it related to different performance of the gridded precipitation as a function of elevation? The challenge of estimated accurate precipitation at high-elevation in mountainous terrain when very few stations are available is never clearly discussed in the paper whereas it is a key element when modeling snowpack in these regions.

*Author response: The discussion (and the conclusions) have been rewritten to reflect the new results from AROME-Crocus with terrain-adjusted precipitation. More discussion on the reasons why GridObs-Crocus does not represent the spatial*

*distribution of snow cover during the melt season has also been added, as well as discussion about the challenge of estimating precipitation at high elevations where there are very few stations available.*

**Specific comments**

4. P 3 L6 : what is the meaning of "internal structure"? FSM and JULES do not include a description of the snow metamorphism contrary to Crocus and SNOWPACK.

   *Author response: By writing "internal structure" we mean that the model simulates several snow layers (not just single-layer snow model), of which the individual layers are described by various snow properties (temperature, density, water content etc), but off course the models vary highly in complexity.*

   *New sentence in the paper:*
   *SNOWPACK (Bartelt and Lehning, 2002; Lehning et al., 2002), SURFEX/ISBA/Crocus (Vion35 net et al., 2012) and JULES (Best et al., 2011) are examples of models with multi-layer snow schemes of different complexity 2 aiming to simulate the surface energy balance and the internal layering of the snowpack.*

5. P 3 L 23: the snow maps generated by Vionnet et al (2016) and Quéno et al (2016) cannot be qualified of "snow maps of high quality". Indeed, these two studies have shown the potential of high-resolution NWP system to drive snowpack model in alpine regions but the snow depth bias in their experiments are still large. These studies have also shown that a distributed precipitation analysis system using the guess from the high-resolution NWP system is required.

   *Author response: We have removed the words "high quality".*

6. P 5 L 1-2: the description of the configuration of the soil model is not clear. Do the author mean that the number of soil layers used to solve the evolution of temperature and the soil moisture is different?

   *Author response: We agree that our description of the ISBA-DIFF model was not clear. The force restore with 3 soil layers is run in the operational AROME-MetCoOP system, while in our offline SUREX simulations we use the multi-layer ISBA-DIF soil scheme, always with 14 layers. We have rewritten this section.*

   *New text in paper:*
   *The model used in this study is the detailed snowpack model Crocus (Brun et al., 1992; Vionnet et al., 2012) coupled with the ISBA land surface model within the SURFEX (Surface Externaliseé) interface (Masson et al., 2013). We applied the ISBA-DIF multi-layer soil scheme (Boone et al., 2000; Habets et al., 2003), which uses a diffusive approach for modeling the heat and moisture transport in the soil. The soil was divided into 14 layers, of which the thickness of the individual layers increases with the soil depth. The bottom of the lowest layer was 12 m. The HSWD*

*(Harmonized World Soil Database) 1 km resolution database for soil texture (FAO/IIASA/ISRIC/ISS-CAS/JRC, 2012) was used for the soil properties.*

7. P 5 L 11-13: a spin up of the land-surface scheme is generally required to generate the initial soil conditions (temperature and moisture). Here, the authors uses the default values in SURFEX. They should include a comment on this topic and mention the potential impact at the beginning of the snow season when the snow cover is potentially sensitive to the heat flux from the ground.

*Author response: Thank you for this comment. We have improved our description of the model setup for the soil layers, and the uncertainties related to this.*

*New text in paper:*
*In this study, we start a new simulation on 1 September, with no snow present, and with default values for soil properties, for both 2014/2015 and 2015/2016. The default soil temperature is 11.9 ∘C for the uppermost surface soil layer for 0 m.a.s.l. (sea-level height). The soil temperature is reduced with increasing terrain elevation using a lapse rate of 0.65 ∘C per 100 m, leading to a surface soil temperature of 1.3 ∘C at 1000 m.a.s.l.. We estimate these surface soil values to be representative of the September climate in our study area. Higher temperatures in the deepest soil layers may however represent an uncertain heat contribution for the snow modeling. This effect should be similar for all the experiments though, since the initialization is the same.*

8. P 7 L 25-26 is there a reference that describes the method used to adjust AROME wind speed? Furthermore, the authors should explain how the 500-m temperature product is interpolated on the 1-km grid used for the snowpack simulation.

*Author response: These post-processed AROME variables are unfortunately not published in peer-reviewed journals. However, there is publicly available documentation about these data that is distributed freely by MET Norway. See:*
*https://drive.google.com/file/d/0B-SaEtrDE91WWEJoNkJiUm5TNzg/view*
*This document describes MetCoOp Ensemble Prediction System (MEPS) and also post-processed variables, including wind and temperature used in our experiments. These products are developed for improving the weather forecasts and are produced operationally and published at the national weather forecasting website YR https://www.yr.no/.*

*The wind speed variable is statistically post processed to represent maximum wind speed 10m the last hour. Correction factors are applied to the wind speed depending on wind direction and region.*

*We have added a reference to this documentation. Further questions about these products may be directed to MET Norway.*

*New text in paper:*

*For temperature and wind speed we used statistically post-processed AROME-MetCoOp forecasts to force SURFEX/Crocus, described by Køltzow (2017). These post-processed weather variables are produced operationally by MET Norway for the weather forecast website YR (https://www.yr.no/). The temperature grid has a spatial resolution of 500 m, and is produced using a Kalman filter correction at observation stations (Homleid, 1995). Horizontal interpolation is carried out using decreasing weights with increasing distance from the station. The temperature is further corrected for terrain elevation, which also takes into account vertical temperature profiles in inversion situations in winter time. The AROME-MetCoOp wind speed was statistically post-processed to represent the maximum wind speed at 10 m during the last hour. In addition, correction factors are applied to the wind speed depending on wind direction and region (Køltzow, 2017). The other variables from the raw AROME-MetCoOp 2.5 km forecasts were interpolated to 1 km spatial resolution using bilinear interpolation, in order to combine the meteorological forecasts with the gridded observations (with a spatial resolution of 1 km) and to run the SURFEX/Crocus model with 1 km grid spacing. The 500 m post-processed AROME-MetCoOp temperature (Køltzow, 2017) wasnalso interpolated by a bilinear method to 1 km resolution. The spatial interpolation was carried out using the File Interpolation, Manipulation and EXtraction library (http://fimex.met.no).*

9. P 17 L 10-15: it is surprising to see that the authors have selected the simulation GridObs-Crocus at Midstova to test the impact of blowing snow sublimation. Indeed, at this station, the quality of the precipitation forcing is known to be bad for the beginning of Winter 2014/2015, leading to a underestimation of snow depth at the beginning of the season. It would have been more interesting to test the impact of the parameterization of blowing snow sublimation using the AROME forcing. Brun et al (2013) have tested the parameterization of blowing snow sublimation in a different context and have shown there may be a redundancy between using a corrected precipitation dataset and accounting for blowing snow sublimation (see Sect. 5 of Brun et al., 2013).
Brun, E., Vionnet, V., Boone, A., Decharme, B., Peings, Y., Valette, R., ... & Morin, S. (2013). Simulation of northern Eurasian local snow depth, mass, and density using a detailed snowpack model and meteorological reanalyses. Journal of Hydrometeorology, 14(1), 203-219.

*Author response: Thank you for this suggestion. We have now followed your recommendations and used AROME forcing for the 1D experiment instead of GridObs forcing. We have added a new figure 8 to the revised paper. The new figure shows the same pattern as the previous one: slightly reduced snow depth for both stations and both years, which improves the bias for AROME-Crocus.*

*New text in paper:*
*In this experiment, SURFEX/Crocus was run twice in 1D mode for these 2 locations: one experiment with identical settings as AROME-Crocus, and one nearly identical with the exception of the option for sublimation in case of snowdrift (AROME-Crocus+BS). The results are shown in Fig. 8. For both locations, the snow*

*depth in AROME-Crocus+BS is decreased, as expected. For both locations, this is an improvement. The bias in AROME-Crocus was +13 cm for Hemsedal II, which has improved to +5 cm in AROME-Crocus+BS. For Midtstova, AROME-Crocus significantly overestimates the snow depth (bias: +104 cm), this is improved in the AROME-Crocus+BS experiment (+83 cm), although the overestimation is still very large. The length of the snow season is reduced by a few days for both stations and both years, similar to the results found in Brun et al. (2013). However, for Midtstova, using blowing snow sublimation does not improve the AROME-Crocus experiment to the extent that it performs equally well as GridObs-Crocus.*

10. P 23 L 15-19 : here the argumentation made by the authors can be discussed. Personally, I always think that model evaluation should be carried out at stations where we know that the model is supposed to represent the meteorological conditions at this station. The model can be right for wrong reasons at these stations. It does not mean that these stations must be included in the evaluation.

   *Author response: We understand your viewpoint, and if we would have more stations available this would be a good solution. With the limitations of available stations in our study area, we chose not to exclude any stations.*

   *However, while working on the revised paper we have realized that the orography discussed in the paper was not the one actually used in SURFEX/Crocus, as it was overwritten by the orography from the climatology files (which use GTOPO30 orography from the U.S. Geological Survey and has a grid spacing of ~1 km) after the forcing was generated. We have also updated figures 2 and 11 and changed the elevation mentioned in the titles (in brackets) of figure 4. The orography used in SURFEX/Crocus is closer to the actual height of the stations than we previously presumed.*

11. P 26 L 8-14: the authors explain that snow melt is underestimated in both experiments but still the snow cover melts too fast in GridObs-Crocus. As mentioned above, this suggest an underestimation of snow accumulation in GridObs-Crocus that could be better discussed in the paper.

   *Author response: There is indeed a negative bias in the gridded precipitation used in GridObs-Crocus, especially for data-sparse areas (e.g. high-mountainous areas) and for intense precipitation. Missing episodes of intense snowfall would explain part of the underestimation of the snow depth in GridObs-Crocus, and large parts of our domain are in high-mountainous areas. This is mentioned at the start of 4.2 (Quality of the forcing data sets) in the discussion. We have put more emphasis on this issue in the discussion and conclusion, see also our reply to comment 3 above and to comment 4 from the editor (below).*

**Technical comments**

12. P3 L 4: "SnowPack" is usually written "SNOWPACK" when referring to the Swiss snowpack model.

    *Author response: We have replaced SnowPack with SNOWPACK.*

13. P 3 L 30: "??" appear at many places in the text and should be corrected.

    *Author response: We cannot find any "??" at page 3, line 30 or anywhere else in the paper. Not in the pdf that was submitted and neither in the pdf with edits from the editor. Could it somehow be a problem with how the file shows on the reviewers side?*

14. P 6 L 3: km2 instead of km-2

    *Author response: Fixed, thank you.*

15. P 22 L 5: do the authors mean "too late" for AROME Crocus and "too early for GridObs-Crocus?*

    *Author response: Correct, this was the wrong way around by mistake. It has been fixed now. Thank you for noticing this!*

16. P 25 L 32: mention explicitly "wind speed" and "air temperature" as the post-processed variables.

    *Author response: We assume you mean page 24 line 30. The sentence has been changed to explicitly mention the post-processed variables.*

**Review by editor**

*Thank you very much for reviewing our paper. Your suggestions very much improved our manuscript. Like you suggested, we will not discuss all the edits you proposed in the pdf document. We have made changes to our manuscript based on nearly all of your comments. Below we will answer to those comments that we think require further explanation.*

1. On MODIS snow cover images: The NOAA-IMS 1-km product is available from 2015 and provides continuous daily coverage based on manual interpretation of multiple sources of snow cover information.

   *Author response: Thank you for your suggestion. We used a different dataset, which was processed and made available by NVE (the Norwegian Water Resources and Energy Directorate) for our project. These data were processed for a period until 2015, and only for cloud-free conditions. This is why we focus on the first season for the SCA analysis.*

2. On defining melting snow days: This analysis would be more convincing comparing simulated snow melt with observed runoff.

   *Author response: This would be an interesting addition, but we think the paper should focus on snow simulations. Inclusion of a more hydrological viewpoint would change the scope of this paper. It would also be difficult to compare directly, as observations of runoff are only available for a few catchment areas, and are representative for a larger area while we focused on grid points. For GridObs-Crocus, the results have been compared at catchment level and to runoff observations in the following paper (accepted for publication 3 May 2018):*

   *Skaugen, T., Luijting, H., Vikhamar-Schuler, D., Müller, K., Stranden, H., and Saloranta, T.: In search of operational snow model structures for the future - comparing four snow models for 17 catchments in Norway, Hydrology Research, 2018 (accepted May 2018)*

3. About the conclusion: Snow melt is underestimated in both experiments: Not a strong part of the analysis.

   *Author response: We agree, and we have removed this item from the conclusions.*

4. So how come Crocus gets good snow depth simulations from a precip dataset that is not corrected for undercatch? Was precip adjusted upward in the model run table? Or is snow density underestimated by Crocus?

   *Author response: The question about correcting for undercatch of precipitation has been much discussed regarding the operational snow simulations carried out by MET-Norway and NVE. The gridObs-dataset is produced for the seNorge snow*

*simulations ([www.seNorge.no](www.seNorge.no)). There have been developed several versions of the seNorge precipitation 1 km grids. The first version (seNorge v1.0) was based on interpolation of precipitation measured at stations, which included a correction factor for precipitation due to undercatch (Lussana et al. 2018). However, these correction factors were found to be too high in the mountains, giving too much precipitation and snow simulations were too high. Stations were classified based on empirical evaluation of whether a station was exposed or not (altitude, terrain exposition etc). In the seNorge v.2.0 it was decided to remove all undercatch correction values, and interpolate based on uncorrected precipitation. This is the reason why all seNorge precipitation grids, including the GridObs dataset do not include correction for undercatch of precipitation. There are specific projects going on at Haukeliseter test area, South Norway, to define improved functions for undercatch, based on observations of wind, precipitation phase, temperature etc. Based on the experiences by the operational snow services, we therefore use the same type of dataset as used in operational snow service ([www.seNorge.no](www.seNorge.no)), until there have been developed undercatch correction values that are really representative for all weather types, and the various stations. Wind is causing errors, both for measuring precipitation, but another effect is that redistribution of snow due to wind is not really included in the snow models. Within a 1 km grid cell we assume that the snow blows inside the cell. A single station represents one location site, and may of course not always be representative of the snow depth within the entire 1 km cell, particularly in mountainous areas, above the tree line, where there often are strong winds. Development of good correction functions for undercatch in mountainous environments is a topic for further research and analysis.*

New paragraph added to Section 2.3.2 Gridded observations:

[revised manuscript text omitted]

---

## Author Response (AR3)

**Report #1 - Anonymous Referee #2**

*Note: Reviewer comments are in black, author responses are in blue Italic.*

General comments

During the additional round of review, the authors have made several changes to the first revised version of the manuscript including:

- A new Crocus simulation driven by AROME-MetCoop including a modified precipitation phase based on the high-resolution post-processed temperature data. This new modification brings large improvements in terms of error statistics for the simulation of snow depth with Crocus driven by AROME-MetCoop
- Large parts of the conclusion and the discussion have been rewritten accordingly.

These changes have improved the quality of the manuscript and the new simulation nicely illustrates the necessity to include an adjustment of precipitation phase in complex terrain when carrying a simulation with a snowpack scheme running at higher resolution that the original atmospheric forcing. Based on this modification, I recommend this paper to be accepted for publication in TC. I mentioned below a few specific comments that the authors should take into account prior to publication.

*Author response: Thank you very much for reviewing our paper again. We are very grateful for all your comments and suggestions during this process, and we are very happy that our paper is now accepted with minor revisions. We will reply to your specific comments below, which we have numbered for easier reference.*

*Thanks again for all your help along the way.*
*Kind regards, on behalf of all authors,*
*Hanneke Luijting*

**Specific comments**

Note: The numbering of the lines is based on the revised version of the paper in track-change mode

1. P4 L21: Masson et al (2013) would be a more appropriate reference for the list of different snowpack schemes in SURFEX.

   *Author response: We have added the reference to Masson et al (2013) where we mention the three snow schemes available in SURFEX (P4, L13 and L19). We have also added a reference to the D95 snow scheme (Douville et al, 1995).*

2. P4 L30: add precipitation phase in the list of weather variables.
   *Author response: We have added precipitation phase in the list of weather variables. New sentence: "Combining observations and NWP data for important weather variables (temperature, precipitation and precipitation phase) as driving data for the snow simulations should better represent the actual observed weather conditions."*

3. P 14 Figure 4: the quality of the fonts and the lines on this figure is poor compared to other figures in the paper such as Fig. 3 or Fig. 8 (which is very similar to Fig. 4). It would be good to produce a new figure easier to read.

   *Author response: Figure 4 has been improved, the better quality of the lines and the fonts makes it easier to read now. Thank you for your suggestion.*

4. P26–P27: the comparison between the simulations with and without the correction of precipitation phase is quite interesting. Could the authors includes the results in terms of bias and RMSE on snow depth to quantify the improvement obtained when considering the adjustment of precipitation phase? I guess these values correspond to the ones removed from Table 2 and 3.

   *Author response: We have added the following sentence to section 4.2: "The bias in snow depth improved from +42 cm (using raw AROME-MetCoOp precipitation) to +20 cm (using the terrain-adjusted AROME-MetCoOp precipitation), and the RMSE improved from 68 cm (raw) to 56 cm (terrain-adjusted)."*

5. P 30 L6: the term "significant" may not be appropriate since the authors did not the statistical significance of the differences between the 2 simulations.

   *Author response: You are right. We have changed the sentence to read "This is an improvement as it reduces the overestimation by AROME-Crocus, but only to some degree."*

6. Conclusion P31 L 4-9: for the results obtained with GridObs-Crocus, it would be good to make the clear distinction between (i) the underestimation of snow accumulation at high-elevation and (ii) the tendency of the model to underestimate snow melt. According to me, the sentence "during the melting season the snow cover melted away too fast" can suggest that snow melt is overestimated in GridObs-Crocus which leads to a snow cover that vanish too early. The authors should consider rewriting this sentence. The same comments apply to the Abstract (P 1, L15-20) and to the discussion (P25, L 4-14).

   *Author response: you are right, this could be misinterpreted We have rewritten this conclusion bullet point to:*

[revised manuscript text omitted]